# Benefits and Pitfalls of Reinforcement Learning for Language Model Planning: A Theoretical Perspective

**Siwei Wang**[1†], **Yifei Shen**[1†], **Haoran Sun**[2†], **Shi Feng**[3†], **Shang-Hua Teng**[4], **Li Dong**[1], **Yaru Hao**[1], **Wei Chen**[1✉]

[1]Microsoft Research Asia, [2]Peking University, [3]Harvard University, [4]University of Southern California

## Abstract

Recent reinforcement learning (RL) methods have substantially enhanced the planning capabilities of Large Language Models (LLMs), yet the theoretical basis for their effectiveness remains elusive. In this work, we investigate RL's benefits and limitations through a tractable graph-based abstraction, focusing on policy gradient (PG) and Q-learning methods. Our theoretical analyses reveal that supervised fine-tuning (SFT) may introduce co-occurrence-based spurious solutions, whereas RL achieves correct planning primarily through exploration, underscoring exploration's role in enabling better generalization. However, we also show that PG suffers from diversity collapse, where output diversity decreases during training and persists even after perfect accuracy is attained. By contrast, Q-learning provides two key advantages: off-policy learning and diversity preservation at convergence. We further demonstrate that careful reward design is necessary to prevent Q-value bias in Q-learning. Finally, applying our framework to the real-world planning benchmark Blocksworld, we confirm that these behaviors manifest in practice.

## 1 Introduction

Planning is a fundamental cognitive construct that underpins human intelligence, shaping our ability to organize tasks, coordinate activities, and formulate complex solutions such as mathematical proofs. It enables humans to decompose complex goals into manageable steps, anticipate potential challenges, and maintain coherence during problem solving. Similarly, planning plays a pivotal role in state-of-the-art Large Language Models (LLMs), enhancing their ability to address structured and long-horizon tasks with greater accuracy and reliability.

Early generations of LLMs primarily relied on next-token prediction and passive statistical learning, which limited their planning capabilities to short-horizon, reactive responses. The o1 family of models represents a major advance in planning by incorporating reinforcement learning (RL) objectives that reward accurate, multi-step reasoning and penalize errors. Inspired by the success of o1, RL has been applied to enhance planning capabilities in various settings, including task decomposition for tool use (Wu et al., 2024a; Luo et al., 2025) and gaming (Yang et al., 2024), visual-language spatial navigation (Chu et al., 2025), and long-horizon robotics tasks (Dalal et al., 2024). These approaches have demonstrated significantly better performance than their supervised fine-tuning (SFT) counterparts. For more related works, please refer to Appendix B.

Despite recent successes, the theoretical basis underlying RL's advantage over SFT in planning tasks and the limitations of current RL methods remain to be established. To enable a tractable analysis of the gradient dynamics, we adopt the data generation model from (Wang et al., 2024b). Within their framework, planning is abstracted as a path-finding problem over a graph structure. For example, a tool-use scenario can be modeled as identifying a valid call sequence within an API call graph (Wu et al., 2024b).

---

*† denotes equal contribution. ✉Corresponding author (weic@microsoft.com).

To capture the fundamental limitations of SFT in planning, we begin by presenting a structural characterization of its stable point for path planning (**Section 3**). Our analyses, expanding the observation of Wang et al. (2024b) that transformer-based LLM architectures cannot identify reachability relationships through transitivity in SFT, show that it introduces co-occurrence-based spurious solutions into planning tasks. This characterization provides a basis for comparison with and motivation for using the RL-based learning approach in language model planning.

Focusing on the behaviors of RL-based learning dynamics, we first consider policy gradient (PG), a widely adopted algorithm for tuning large language models (**Section 4**). Our analysis yields three key findings. First, with only 0-1 outcome rewards, each iteration of PG equivalently corresponds to an SFT process on the exploration data; however, PG empirically outperforms SFT due to the exploration-driven data augmentation it enables. Second, although PG converges to a model that outputs correct paths for all source–target pairs seen during training, we uncover a diversity collapse phenomenon: the model's output diversity steadily declines throughout training and continues to diminish even after achieving 100% training accuracy. Third, we show that KL regularization acts as an explicit diversity-preserving term, but at the expense of accuracy.

We then analyze Q-learning, a paradigm well known in game playing but rarely applied to LLMs (Mnih et al., 2013) (**Section 5**). Our analysis yields two key findings. First, when trained with only an outcome reward signal, Q-learning suffers from Q-value bias; however, incorporating process rewards eliminates this issue. Second, once this issue is addressed, Q-learning offers two theoretical advantages over PG: it converges to a solution that preserves output diversity when achieving optimal training accuracy, and it naturally supports off-policy learning. The latter is particularly important in practice, since rollouts performed with a quantized model or large batch sizes are effectively off-policy, as exemplified by the VeRL framework (Sheng et al., 2024). Finally, we validate all these theoretical findings through experiments.

To summarize, our main contribution is a theoretical treatment of the impact of reinforcement learning on language model planning. Our mathematical analysis of learning dynamics sheds light on phenomena observed in practice–for example, SFT tends to memorize while RL promotes generalization; PG methods often suffer from diversity collapse; and KL regularization helps mitigate diversity degradation, albeit at the cost of reduced accuracy. Other findings point to promising future directions, such as leveraging Q-learning to achieve both diversity and accuracy, as well as enabling off-policy learning. Taken together, these results provide a principled foundation for understanding and advancing reinforcement learning methods in language model planning.

## 2 PRELIMINARIES

### 2.1 PATH PLANNING DATASET: SYNTAX AND DATA SOURCES

Following (Wang et al., 2024b), we abstract planning in large language models as path planning over an *unknown* directed graph $\mathcal{G} = (\mathcal{V}, \mathcal{E})$, where $\mathcal{V}$ represents the set of nodes and $\mathcal{E}$ represents the set of edges. Each node $v \in \mathcal{V}$ is represented by a unique token. The language model's vocabulary consists of these node tokens and a special end-of-sequence token, \n. An edge $(u, v) \in \mathcal{E}$ signifies a directed connection from node $u$ to node $v$. A node $t$ is reachable from a node $s$ if a directed path from $s$ to $t$ exists in $\mathcal{G}$. We denote by $\mathbf{A} \in \{0, 1\}^{|\mathcal{V}| \times |\mathcal{V}|}$ the adjacency matrix of $\mathcal{G}$, where $\mathbf{A}[u, v] = 1$ if and only if $(u, v) \in \mathcal{E}$, and by $\mathbf{R} \in \{0, 1\}^{|\mathcal{V}| \times |\mathcal{V}|}$ the reachability matrix, where $\mathbf{R}[t, s] = 1$ if and only if $t$ is reachable from $s$.

**Running Example (Blocksworld).** To connect this abstraction to real-world LLM planning scenarios, consider the Blocksworld domain (Valmeekam et al., 2023b). In Blocksworld, we are given several colored blocks (e.g., Grey, Black, Red, White) placed either on a table or stacked on each other, and the task is to transform an initial arrangement (*source state*) into a target arrangement (*target state*) using valid moves. For example, the *source state* may place all blocks on the table, and the *target state* requires stacking them so that Red is on Grey, Grey is on Black, and Black is on White. We map every distinct block configuration to a node in $\mathcal{V}$; edges in $\mathcal{E}$ correspond to valid single moves such as "place White on Grey". A valid plan is therefore equivalent to a path in $\mathcal{G}$ connecting the source node $s$ and target node $t$.

This abstraction matches natural language planning tasks: in the original benchmark, the LLM is given two textual descriptions of initial and target states and asked to generate a sequence of natural language actions to achieve the goal. In our abstract setup, we strip away language semantics to focus on the core planning structure while retaining the same problem difficulty.

The set of all reachable source-target pairs $(s, t)$ is partitioned into a training set $D_{\text{Train}}$ and a test set $D_{\text{Test}}$. We define three corresponding data stages:

- **SFT Training Data:** We construct a training dataset $\mathcal{D}^{\text{SFT}}$ for supervised fine-tuning by sampling multiple $(K)$ paths for each reachable pair $(s, t) \in D_{\text{Train}}$ by random walk. Each training data in $\mathcal{D}^{\text{SFT}}$ is a sequence in the format "$s\ t\ s\ a\ b\ c\ t$ \n", where $s\ a\ b\ c\ t$ are tokens for nodes in a valid path from $s$ to $t$, and \n indicates the end of the sequence. We call the model after SFT training the *base model*.

- **RL Training Data:** We sample pairs $(s, t)$ from $D_{\text{Train}}$ and let the model itself (on-policy) or the base model (off-policy) generate the remaining tokens in the sequence. When the model outputs \n or the generation reaches the maximum length, an outcome reward or some step rewards will be given, depending on the used reward format.

- **Test Data:** When testing, we provide pairs $(s, t)$ from $D_{\text{Test}}$, which are never encountered in either SFT or RL training. The model is tasked with generating a valid path from $s$ to $t$.

Throughout the empirical study, we use a one-layer, single-head Transformer as the backbone model. The embedding size is set to $d = 120$. The graph $\mathcal{G}$ in our main empirical validation is generated using the Erdős-Rényi model with $|\mathcal{V}| = 100$ nodes and an edge probability of $0.15$. The ratio of the sets $|D_{\text{Train}}|/|D_{\text{Test}}|$ is approximately $0.25$ (approximately 20% pairs are in $D_{\text{Train}}$). The number of paths sampled for each reachable pair in $D_{\text{Train}}$ is $K = 10$. We also consider the graph $\mathcal{G}^{BW}$ that characterizes the transition between different block configurations in Blocksworld, which is proposed by Valmeekam et al. (2023a) to evaluate the LLM's planning ability. The details for the graph construction are presented in Appendix G.

## 2.2 REINFORCEMENT LEARNING ALGORITHMS

We first define the notation. Given a vector $\mathbf{x}$, we denote its $m$-th element by $\mathbf{x}[m]$. For a given sequence "$s\ t\ s\ a\ b\ c\ t$ \n", we represent it as $\mathbf{u} = (u_{\text{source}}, u_{\text{target}}, u_1, \cdots, u_m, \cdots)$. We denote by $\hat{\boldsymbol{u}}_m$ the output probability vector of the current model at the $m$-th position, and by $\hat{\boldsymbol{u}}_m^{\text{base}}$ that of the base model before RL. The model parameters are denoted by $\theta$.

**Policy Gradient.** Let $\mathcal{P}$ be the set of valid paths. The outcome reward is only given at the end of the path and is defined by $R(\mathbf{u}) = r\,\delta_{\mathbf{u} \in \mathcal{P}} + p$, where $r > 0$ and $p$ are constants, and $\delta$ denotes the indicator function that is $1$ if condition is true and $0$ otherwise. For an individual trajectory, the loss function is

$$\ell = -\sum_{m \geq 1} \left( \underbrace{R(\mathbf{u}) \log \hat{\mathbf{u}}_m[u_{m+1}]}_{\text{Policy Gradient}} + \lambda \underbrace{\log \hat{\mathbf{u}}_m[u_{m+1}] \left\{ \log \frac{\hat{\mathbf{u}}_m[u_{m+1}]}{\hat{\mathbf{u}}_m^{\text{base}}[u_{m+1}]} \right\}}_{\text{KL Divergence}} \right), \qquad (1)$$

where $\lambda$ controls the KL regularization strength, and $\{\cdot\}$ means the term is detached and will not contribute to the gradient.

**Q-Learning.** The goal of Q-learning is to approximate the Q-function with the model logits. Let $Q_\theta(s_m, a_m)$ be the Q-function where $s_m = (u_{\text{source}}, u_{\text{target}}, u_1, \cdots, u_m)$ is the state, $a_m \in \mathcal{V}$ is the action, and $s'_m = (u_{\text{source}}, u_{\text{target}}, u_1, \cdots, u_m, a_m)$ is their next state. The objective is $\sum_m \left( Q_\theta(s_m, a_m) - [R(s_m, a_m) + \max_{a'_m} Q_\theta(s'_m, a'_m)] \right)^2$. We denote the logits at step $m$ by $\tilde{\mathbf{u}}_m$. For an individual trajectory, the loss is given by

$$\ell = \sum_{m \geq 1} \left( \tilde{\mathbf{u}}_m[u_{m+1}] - R(\mathbf{u}, m) - \left\{ \max_k \tilde{\mathbf{u}}_{m+1}[k] \right\} \right)^2. \qquad (2)$$

For Q-learning's reward $R(\mathbf{u}, m)$, we study two scenarios: (i) *outcome reward*, where the reward depends on whether the path is correct, and (ii) *process reward*, where intermediate rewards are given based on adjacency and target checks. Specifically,

$$R(\mathbf{u}, m) = \begin{cases} \delta_{\mathbf{u} \in \mathcal{P}} \delta_{u_{m+1} = u_{\text{target}}}, & \text{If outcome reward,} \\ \underbrace{\delta_{u_{m+1} = u_{\text{target}}}}_{\text{Target check}} - \underbrace{\delta_{(u_m, u_{m+1}) \notin \mathcal{E}}}_{\text{Adjacency check}}, & \text{If process reward.} \end{cases} \tag{3}$$

That is, in the outcome reward setting, a reward of 1 is given only if the entire path is valid, and it is assigned at the step when the target is reached. In contrast, in the process reward setting, we do not check whether the entire path is valid or not. The model is always rewarded upon reaching the target, but it is also penalized at any step that transitions to a non-adjacent node.

## 3 LIMITATIONS OF SUPERVISED FINE-TUNING IN PLANNING

Focusing on the stationarity of the training dynamics, we present a basic structural characterization that captures a fundamental limitation of SFT in planning. Our analysis builds on an early finding of Wang et al. (2024b), which showed that transformer-based SFT planning approaches lack transitivity-learning mechanisms needed to obtain complete reachability structures. The new characterization expands and complements the earlier results and provides a theoretical explanation for why SFT-based planning tends to rely on memorization. More importantly, this result establishes a theoretical basis for comparison with RL-based planning frameworks and highlights the role of exploration in achieving better generalization during the adaptive learning process.

### 3.1 DISCUSSIONS ON EXISTING FINDINGS

To set up our characterization, we first review the analysis framework of Wang et al. (2024b), which examines the training dynamics of a one-layer, single-head Transformer under an autoregressive loss function. Their analysis shows that, during training, the model encodes both the adjacency and reachability structures of the underlying graph in its learnable parameters. The model then predicts the next node in a sequence by ensuring that it is adjacent to the current node and lies along a path toward the target node. A full description of their approach is given in Algorithm 1 in Appendix C.

Wang et al. (2024b) showed, both theoretically and experimentally, that the adjacency and reachability information stored in a model's weights is generally **incomplete**. To formalize this, consider a training dataset $\mathcal{D}^{\text{SFT}}$. The *observed adjacency matrix* $\boldsymbol{A}^{\text{obs}}(\mathcal{D}^{\text{SFT}})$ contains exactly those edges $(j, k)$ that appear in at least one path from $\mathcal{D}^{\text{SFT}}$. Similarly, the *observed reachability matrix* $\boldsymbol{R}^{\text{obs}}(\mathcal{D}^{\text{SFT}})$ records that a target node $t$ is reachable from an intermediate node $k$ if $\mathcal{D}^{\text{SFT}}$ contains a sequence with target $t$ in which $k$ occurs as a non-source node. We refer to such pairs $(t, k)$ as *observed reachable pairs*.

However, we find that even when an adjacency relation appears in $\mathcal{D}^{\text{SFT}}$, the SFT model may not learn a high weight for it. To illustrate this, we run experiments on the Blockworld dataset, and the results are presented in Figure 1. In Figure 1a, we show the frequency of all adjacency relationships in the training set (every adjacency relationship appears at least once), where brighter regions indicate higher frequencies. Then Figure 1b displays the corresponding weights learned after SFT. By comparing them, we observe that some adjacency relationships present in the data are not well captured by the model, especially those with low frequency. This observation motivates us to further investigate the model's stable (optimal) points.

### 3.2 CHARACTERIZATION OF THE STABLE POINT IN SFT-BASED LEARNING DYNAMICS

Building on the observation of Wang et al. (2024b) that next-node prediction depends mainly on the current and target nodes, we adopt the following natural assumption about model expressiveness for our structural characterization. Recall that $u_{\text{target}}$ and $u_m$ denote the target node and the current node at position $m$, respectively.

**Assumption 3.1.** *The model's predicted logits for the next token can be expressed as a function of the (target, current) node pair, i.e., there exists a function $\mathbf{f}$ such that the logits $\tilde{\mathbf{u}}_m = \mathbf{f}(u_{target}, u_m)$.*

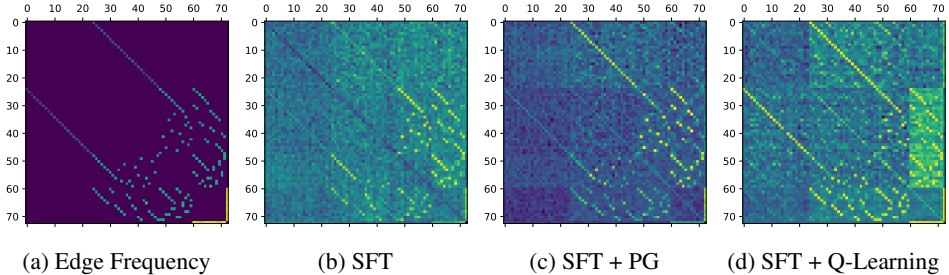

Figure 1: Frequency of edge occurrences in the SFT training data $\mathcal{D}^{\text{SFT}}$ and the adjacency structures learned by different models. The underlying graph represents transitions between block configurations in Blocksworld (Valmeekam et al., 2023a).

Note that in the assumption, $\mathbf{f}$ can be an arbitrary function. Our experiments validate this assumption. As shown in Section G.1, the evolution of attention maps during training for SFT, PG, and Q-learning demonstrates that the trained transformer acts primarily as a function of the target and current nodes.

We now characterize the structure of the stable point achieved by SFT. Due to space limitations, we defer all the proofs in this paper to the appendix.

**Theorem 3.1** (Optimal Solution of SFT). *Assume Assumption 3.1 holds. Let $N_{u_{target},u_m,k}$ denote the number of occurrences in the training dataset where the target node is $u_{target}$, the current node is $u_m$, and the next node is $k$. The optimal solution of SFT satisfies:*

$$\frac{\exp(\mathbf{f}(u_{target}, u_m)[k])}{\sum_{k'} \exp(\mathbf{f}(u_{target}, u_m)[k'])} = \frac{N_{u_{target},u_m,k}}{\sum_{k'} N_{u_{target},u_m,k'}} \quad if \sum_{k'} N_{u_{target},u_m,k'} > 0.$$

*If $\sum_{k'} N_{u_{target},u_m,k'} = 0$, output can be any probability distribution.*

**Takeaway 1: SFT memorizes co-occurrence relationships in the training dataset.**

Theorem 3.1 extends the findings of Wang et al. (2024b), which showed that SFT-based mechanisms may fail to learn the complete adjacency and reachability matrices, leading to spurious correlations. However, those earlier results did not specify the nature of the solutions to which the model converges. Complementing their work, Theorem 3.1 clarifies this by showing that SFT essentially memorizes co-occurrence relationships among the target node, the current node, and the immediate next node based on their frequencies in $\mathcal{D}^{\text{SFT}}$. Hence, SFT will fail to exploit transitivity information (which never appears in $\mathcal{D}^{\text{SFT}}$) to capture the true graph connectivity required for path planning.

In Figure 1, we further compare the weights of models trained by two RL approaches, PG and Q-learning. Both RL approaches capture the adjacency relationships better. Similar findings are reported by Chu et al. (2025), who empirically observe that SFT tends to memorize while RL exhibits better generalization. Our structural analysis in Theorem 3.1 provides a theoretical explanation for the first part of this phenomenon, namely, why "SFT memorizes". In the following sections, we examine the two RL-based approaches, PG and Q-learning, and provide a theoretical explanation of the second part, i.e., why "RL generalizes".

## 4 PATH PLANNING CAPACITIES OF POLICY GRADIENT

In this section, we examine the path-planning capacity of the policy gradient, the core principle behind advanced RL algorithms such as PPO (Schulman et al., 2017) and GRPO (Shao et al., 2024). Understanding the strengths and limitations of the basic policy gradient provides theoretical insights into its behavior, highlights the mechanisms that enable effective path planning, and clarifies the challenges that motivate more sophisticated approaches.

## 4.1 THEORETICAL ANALYSIS

We first establish the connection between policy gradient (PG) and supervised fine-tuning (SFT), highlighting the potential advantages of PG over SFT. We then analyze PG's training dynamics and show that, without KL regularization, the model can achieve 100% training accuracy (under temperature sampling) while progressively losing output diversity. Finally, we demonstrate that, when initialized with a reasonably capable base model, adding a KL regularization helps preserve diversity and thereby enhances generalization, albeit sometimes at the cost of accuracy.

To make this connection precise, we show that the PG loss function closely resembles the SFT loss, restricted to the subset of data generated during RL training that corresponds to correct paths.

**Theorem 4.1** (Connections between PG and SFT). *Assume Assumption 3.1 holds. Let $\mathcal{D}^{RL,t}$ denote the set of data generated during the RL training step $t$. When $r = 1$, $p = 0$ (i.e., reward 1 for a correct path and reward 0 otherwise) and $\lambda = 0$ (i.e., without KL regularization), the loss function of Policy Gradient is the same as the loss function of using SFT only on correct paths in $\mathcal{D}^{RL,t}$.*

As shown by Wang et al. (2024b), SFT can learn the adjacency and reachability relations. Thus, Theorem 4.1 shows that PG can capture these relations presented in the dataset $(\cup_{t=1}^{T} \mathcal{D}^{RL,t}) \cap \mathcal{P}$. However, unlike SFT, which relies on a fixed training dataset, PG generates data on-policy during training. As the model improves, it can explore and discover new correct paths that were absent from the initial training set. This exploration-driven data augmentation enables PG to achieve stronger performance beyond what SFT alone can provide.

**Takeaway 2: PG outperforms SFT primarily because its iterative data generation process encourages exploration and effectively expands the training dataset.**

Building on the loss function, we analyze the gradient and identify two distinctive properties of on-policy PG updates.

**Theorem 4.2** (Convergence of PG without KL regularization). *Assume Assumption 3.1 holds. For any $i, j$ pair, let $C(i, j)$ denote the set of nodes that can reach $i$ and are adjacent to $j$. The following then holds: If $r = 1$, $p = 0$ and $\lambda = 0$, then* (i) *the gradient $\frac{\partial \ell}{\partial \mathbf{f}(i,j)[k]}$ for $k \notin C(i,j)$ is always positive, and* (ii) *the total sum of gradient $\sum_k \frac{\partial \ell}{\partial \mathbf{f}(i,j)[k]} = 0$.*

Theorem 4.2 shows that the logits $\mathbf{f}(i,j)[k]$ corresponding to incorrect tuples $(i, j, k)$, i.e., cases where node $j$ cannot reach node $i$ through node $k$, will continue to decrease, while some other logits will not converge to $-\infty$. Consequently, under gradient descent, the probability that the model outputs a wrong path in $D_{\text{Train}}$ converges to zero.

Next, we analyze how the model's output diversity evolves. Intuitively, the most diverse model that still achieves 100% accuracy is one that produces a uniform distribution over $C(i, j)$ for each target node $i$ and current node $j$. We now analyze the evolution of the KL divergence between this uniform distribution and the model's output distribution during PG training without KL regularization.

**Theorem 4.3** (Diversity Collapse of PG without KL regularization). *Assume Assumption 3.1 holds. Let $U_{C(i,j)}$ denote the uniform probability distribution on support $C(i, j)$. When $r = 1$, $p = 0$ and $\lambda = 0$, and logits $\mathbf{f}^t(i,j)[k]$ for $k \notin C(i,j)$ is $-\infty$, where $\mathbf{f}^t(i,j)$ denotes the logits value of $\mathbf{f}(i,j)$ at time step $t$. For any such PG gradient descent step $t$, we have that*

$$KL(U_{C(i,j)}||\textbf{softmax}(\mathbf{f}^t(i,j))) \leq \mathbb{E}[KL(U_{C(i,j)}||\textbf{softmax}(\mathbf{f}^{t+1}(i,j)))].$$

Note that the metric $KL(U_{C(i,j)}||\textbf{softmax}(\mathbf{f}^t(i,j)))$ takes minimum value when $\textbf{softmax}(\mathbf{f}^t(i,j))$ is also the uniform distribution on $C(i, j)$, and takes maximum value when $\textbf{softmax}(\mathbf{f}^t(i,j))$ is a one-hot vector. Thus, Theorem 4.3 demonstrates that even after attaining 100% accuracy on $D_{\text{Train}}$, the model continues to exhibit declining output diversity.

**Takeaway 3: In the absence of KL divergence, output diversity continuously declines.**

This diversity-collapse phenomenon has been reported in the literature (Cui et al., 2025) and can impair a model's ability to generalize. To address it, many techniques have been proposed, the most common being KL regularization. To better understand its role, we analyze the stable point of the model under KL regularization, highlighting both its advantages and limitations.

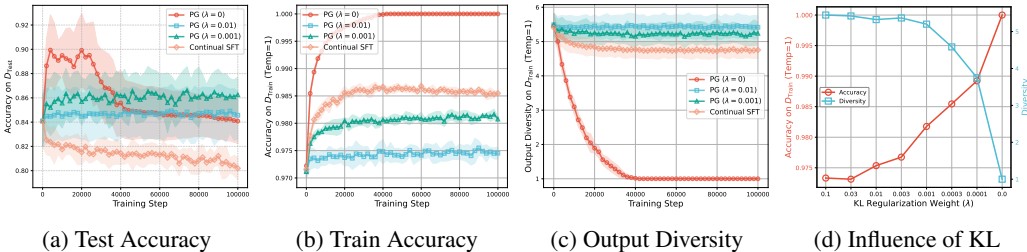

(a) Test Accuracy     (b) Train Accuracy     (c) Output Diversity     (d) Influence of KL

Figure 2: Empirical results of PG training. Both PG and continual SFT are initialized from the same base model. Figures (a)-(c) illustrate the training dynamics of test accuracy (under greedy decoding), training accuracy (under temperature sampling), and response diversity (under temperature sampling). Figure (d) shows how different KL regularization strengths affect the final models.

**Theorem 4.4** (The effect of KL regularization). *When $r = 1$, $p = 0$ and $\lambda > 0$, the stable point of the PG model satisfies the following, under Assumption 3.1: For any fixed $i, j$, either $\mathbf{q}(i, j)[k] = 0$ or $\mathbf{q}(i, j)[k] \propto \mathbf{q}^{base}(i, j)[k] \exp(\mathbf{p}(i, j)[k]/\lambda)$. Here $\mathbf{q}(i, j)[k]$ is the probability of outcome $k$ in $\textbf{softmax}(\mathbf{f}(i, j))$, $\mathbf{q}^{base}(i, j)[k]$ is the probability of outcome $k$ in the base model, and $\mathbf{p}(i, j)[k]$ is the probability of tuple $i, j, k$ belonging to a valid path given output probability $\{\mathbf{q}(i, j)[k]\}_{i,j,k}$.*

This result shows that KL regularization constrains the trained model to remain close to the base model, thereby preserving some of its diversity. This effect is a *double-edged* sword. Consider a valid next node $k$ for which the base model assigns low probability, i.e., $\mathbf{q}^{base}(i, j)[k]$ is small. On the one hand, KL regularization prevents $\mathbf{q}(i, j)[k]$ from becoming arbitrarily small, increasing the chance of generating valid paths involving $k$. On the other hand, it also prevents $\mathbf{q}(i, j)[k]$ from becoming very large, limiting potential gains when the base model's prior is suboptimal. This tradeoff explains seemingly contradictory findings in recent literature: when the base model is already capable, KL regularization preserves diversity and improves generalization, but when the base model is weak, the regularization may hinder learning by overly constraining policy updates.

**Takeaway 4: KL regularization explicitly acts as a diversity-preserving mechanism, provided that the base model is reasonably capable, but this comes at the cost of reduced train accuracy.**

### 4.2 EMPIRICAL VALIDATIONS

The results are presented in Figure 2, where we compare PG with different KL regularization factor $\lambda$ against continual SFT. All models are initialized from the same base model after SFT training, while continual SFT means training the model for more time steps on the same SFT dataset $\mathcal{D}^{\text{SFT}}$. The empirical results match the takeaways we summarized from our theoretical findings, as detailed below.

**Takeaway 2**: In Figure 2a, as the training progresses, the test accuracy of Continual SFT constantly decreases, while all the PG methods can achieve an improvement, since they benefit from exploration-driven training data. **Takeaway 3**: In Figure 2b and 2c, we can see that PG without KL regularization progressively achieves and maintains 100% training accuracy, but its output diversity, i.e., the average number of distinct correct paths generated over 100 sampling trials for the same source-target pair, keeps decreasing during training. In the end, the model eventually produces only one path per pair. Moreover, as shown in Figure 2a, when the diversity diminishes, continued training degrades test accuracy. **Takeaway 4**: As a comparison, PG with KL regularization maintains high output diversity in the end, but their training accuracy is limited. This trade-off is further stated in Figure 2d: with a higher factor $\lambda$, the model can have a higher output diversity and a lower training accuracy. Along with Figure 2a, it is shown that KL regularization prevents the model from deviating too far from the base model in terms of both diversity and training accuracy. This mitigates overfitting but also caps potential gains in test accuracy.

# 5 ANALYSIS OF THE Q-LEARNING-BASED PLANNING MECHANISM

In this section, we analyze the Q-learning mechanism for language-model planning under two different reward designs. We show that stepwise process rewards enable convergence, preserve diversity, and remain valid under off-policy sampling, whereas outcome rewards collapse to trivial solutions. Our analysis begins under Assumption 3.1 for both reward types, and we then extend the process-reward analysis to a more concrete linear Transformer model without this assumption.

## 5.1 THEORETICAL ANALYSIS

To analyze the structure and convergence of the Q-learning stable point, we introduce a mild assumption, which we call the persistent exploration assumption about the RL-based learning dynamics.

**Assumption 5.1** (Persistent exploration). *At training step $t$, let $i_t = u_{target}$, $j_t = u_m$, $k_t$, respectively, denote the target, current, and next nodes. We assume for every $(i, j, k)$, $\exists \underline{N}_{i,j,k}^{\mathrm{prop}} > 0$ such that*

$$\liminf_{T \to \infty} \frac{1}{T} \sum_{t=0}^{T-1} \delta_{(i_t, j_t, k_t) = (i,j,k)} \geq \underline{N}_{i,j,k}^{\mathrm{prop}}.$$

Under the persistent exploration assumption, every coordinate is updated frequently enough to allow convergence analysis. In practice, this assumption is usually satisfied, for instance:

**Lemma 5.1.** *Training with $\epsilon$-exploration (i.e., exploring each alternative action with probability proportional to $\epsilon$) satisfies the persistent exploration assumption.*

With the outcome reward, the signal merely verifies whether the entire sequence constitutes a valid path ending at target $i$. It does not differentiate between current nodes $j$ or candidate next nodes $k$ when $k \neq i$. As a result, at a stable point, all logits collapse to the same constant $c_i$ for each fixed target $i$, causing the parameters to lose structural information, as stated in the theorem below.

**Theorem 5.1** (Stable points of outcome reward). *Assume the RL-training uses the outcome reward $R(\mathbf{u}, m) = \delta_{\mathbf{u} \in \mathcal{P}} \, \delta_{u_{m+1} = u_{target}}$, and a stable point exists under persistent exploration (Assumption 5.1) and Assumption 3.1. Then, at any stable point of the Q-learning model, for each fixed target $i$ and $k \neq i$, all logits $\mathbf{f}(i, j)[k]$ take the same value depending only on $i$.*

With the process reward, the update rule accounts for both adjacency and target conditions. The next theorem establishes that the process converges to well-defined limits that capture the underlying graph structure.

**Theorem 5.2** (Stable points of process reward). *Assume Assumption 3.1 holds and the process reward is used, i.e. $R(\mathbf{u}, m) = \delta_{u_{m+1} = u_{target}} - \delta_{(u_m, u_{m+1}) \notin \mathcal{E}}$. Suppose the score vector $\mathbf{f}(i, j) \in \mathbb{R}^n$ is initialized at zero and updated under the persistent exploration assumption with learning rate $\eta$. Then, in the Q-learning model, as $t \to \infty$, $\mathbf{f}^{(t)}(i, j)[i] \longrightarrow \mathbf{A}[j, i]$, and for $k \neq i$,*

$$\mathbf{f}^{(t)}(i, j)[k] \longrightarrow \begin{cases} 1, & \mathbf{A}[j, k] = 1 \text{ and } \mathbf{R}[i, k] = 1, \\ 0, & \text{exactly one of } (\mathbf{A}[j, k] = 1) \text{ or } (\mathbf{R}[i, k] = 1), \\ -1, & \mathbf{A}[j, k] = 0 \text{ and } \mathbf{R}[i, k] = 0. \end{cases}$$

*Here "$\longrightarrow$" denotes convergence or "tend to". Moreover, the convergence is linear; the effective rate depends on $\eta$ and the update proportions $\underline{N}_{i,j,k}^{\mathrm{prop}}$.*

**Takeaway 5: Different from PG methods, in Q-learning, relying solely on the outcome reward signal can cause Q-value bias, whereas introducing process rewards mitigates this issue.**

To gain further insight in a setting closer to practice, we analyze a simplified but concrete one-layer, single-head linear Transformer without the abstraction of Assumption 3.1.

**Assumption 5.2** (Linear transformer Wang et al. (2024b)). *We work under the simplified Transformer setting in Wang et al. (2024b): (1) The token embedding matrix and the output weight matrix are both set to the identity; (2) Attention is fixed entirely on the target node $u_{target}$, so the attention block contributes only the value lookup $\mathbf{W}^V[u_{target}, \cdot]$; (3) All layer normalizations are removed, and the feedforward block is replaced by a linear map of the form $\mathrm{FFN}(\mathbf{X}) = \mathbf{X}\mathbf{W}^M$. Under these assumptions, the logit decomposes as $\tilde{\mathbf{u}}_{m+1}[k] = \mathbf{W}^M[u_m, k] + \mathbf{W}^V[u_{target}, k]$, where $\mathbf{W}^M$ arises from the feed-forward weights and $\mathbf{W}^V$ from the value matrix of the attention block.*

Despite this simplification, the analyzed results remain consistent with the experiments of real Transformers, as demonstrated in Wang et al. (2024b). This formulation aligns with the actual 1-layer 1-head Transformer architecture, offering greater practical utility compared to the abstract function $\mathbf{f}(i, j)[k]$. The subsequent result characterizes the set of stable points under this decomposition and is consistent with the structural limits established in Theorem 5.2.

**Theorem 5.3** (Stable points of process reward). *Assume Assumption 5.2 holds. For a linear transformer, assume training uses the process reward, and the persistent exploration condition holds. At a stable point of the Q-learning model, for each $k$ there exists $c_k \in \mathbb{R}$ such that*

$$\mathbf{W}^M[j, k] = \mathbf{A}[j, k] - 1 + c_k, \mathbf{W}^V[i, k] = \mathbf{R}[i, k] - c_k.$$

*Conversely, any such $(\mathbf{W}^M, \mathbf{W}^V)$ is a stable point. Hence, the set of stable points is $\{(\mathbf{W}^M, \mathbf{W}^V) : c_k \in \mathbb{R}, \ k \in [|\mathcal{V}|]\}$.*

Theorem 5.1 shows that if only outcome reward is used, the learned logits collapse to a constant across all states for a given target. In contrast, Theorem 5.2 and Theorem 5.3 show that with persistent exploration, process rewards can preserve adjacency and reachability (note that in Theorem 5.3, the constant $c_k$ is immaterial in terms of path planning, since for any stable point, $\tilde{\mathbf{u}}_{m+1}[k] = \mathbf{W}^M[u_m, k] + \mathbf{W}^V[u_{\text{target}}, k] = \mathbf{A}[j, k] + \mathbf{R}[i, k] - 1$, which is the same as Theorem 5.2). Moreover, the convergence holds even under off-policy sampling, and action diversity is preserved because all feasible next nodes converge to the logit value 1.

**Takeaway 6: Compared to PG methods, Q-learning can operate off-policy and better maintains output diversity.**

## 5.2 EMPIRICAL VALIDATIONS

We first examine the training and test accuracy results in Figure 3a, where we compare Q-learning under different reward designs and sampling policies. All models are initialized from the same base model. Figure 3b states the diversity-accuracy trade-off of Q-learning models, policy gradient models, and the continual SFT model (under different temperatures). Figure 4 illustrates the logits of an on-policy Q-learning model with process rewards and fixed attention (attention fixed on the target node $u_{\text{target}}$). The model is initialized from the same base model as in Figure 3, and is further trained with reinforcement steps on all pairs $(s, t) \in \mathcal{D}^{\text{SFT}}$ where $t \in [20]$. In each row $i$, we plot the logits for nodes 0–20 (normalized to $[0, 1]$) when the current node is 0 and the target node is $i$. White indicates larger logits, black indicates smaller logits, and green frames highlight nodes that are both children of node 0 and ancestors of $i$, corresponding to valid outputs. The empirical results are consistent with the takeaways introduced above, as detailed below.

**Takeaway 5:** In Figure 3a, Q-learning with process rewards achieves comparable training accuracy and significantly better test accuracy than the PG model, while Q-learning with outcome rewards collapses and converges to near-zero accuracy on both training and test sets. Examining each row of Figure 4, we observe that the logits of feasible nodes gradually increase and converge to the largest values within their respective rows, which aligns with Theorem 5.2 and 5.3 and confirms that process rewards enable the model to recover the correct graph structure. **Takeaway 6:** In Figure 3a, off-policy Q-learning with process rewards attains training and test accuracy comparable to on-policy Q-learning with process rewards, demonstrating that Q-learning can operate off-policy. Finally, Figure 3b further highlights that the Q-learning process rewards preserve output diversity. Figure 4 also reflects this phenomenon: within each row, the logits of feasible nodes become increasingly close to one another (approaching white) over time, indicating convergence to diverse but correct transitions.

## 6 CONCLUSION

In this paper, we analyze the benefits and limitations of reinforcement learning in language model planning through the lens of learning dynamics. Our theoretical analysis shows that supervised fine-tuning introduces spurious co-occurrence solutions, while policy gradient and Q-learning outperform SFT primarily through exploration. We further identify a critical drawback of basic policy gradient—*diversity collapse*—and show that Q-learning mitigates this issue while supporting off-policy learning. These insights clarify the mechanisms behind the recent success of RL-based

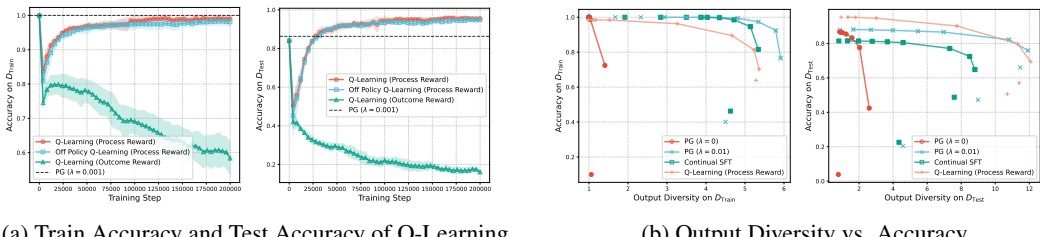

(a) Train Accuracy and Test Accuracy of Q-Learning

(b) Output Diversity vs. Accuracy

Figure 3: Empirical comparison between Q-learning and PG. Figure (a) shows the training dynamics of training and test accuracy (under greedy decoding). Figure (b) compares the Pareto frontiers of output diversity and accuracy on the training and test sets (under temperature decoding).

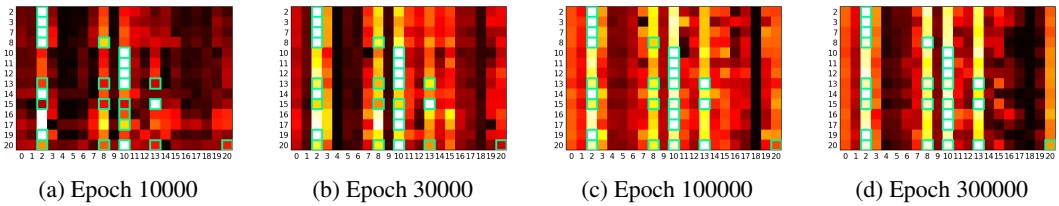

(a) Epoch 10000          (b) Epoch 30000          (c) Epoch 100000          (d) Epoch 300000

Figure 4: Heatmap of normalized logits from the Q-learning model with process reward. For each row $i$, green blocks indicate valid next nodes given the current node $0$ and target node $i$. The logits corresponding to these valid actions consistently increase during training.

approaches and highlight principled research directions, such as leveraging Q-learning for robust, scalable, and generalizable planning in language models.

## REPRODUCIBILITY STATEMENT

We attach all the source code necessary to reproduce our experimental results in the supplementary materials. As for the theoretical results, complete proofs of all the theorems are provided in Appendix C, D, and E for full transparency and verification.

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

## A    THE USE OF LARGE LANGUAGE MODELS

In this paper, the core conceptual framework and its iterative development were driven by human researchers. LLMs served strictly in a supporting capacity, primarily employed for linguistic refinement of the manuscript to enhance readability while preserving original technical content. The whole paper is carefully supervised, reviewed, and modified by the authors who maintain complete responsibility for the scientific validity, technical accuracy, and ethical integrity of this work.

## B    MORE RELATED WORKS

### B.1    PLANNING OF LLMS

Planning is a fundamental component of human intelligence and autonomous agents. Several studies have evaluated the planning capabilities of LLMs trained without reinforcement learning, such as CogEval (Momennejad et al., 2023) and Blockworlds (Valmeekam et al., 2023b). These works consistently report negative results, suggesting that LLMs lack inherent planning abilities. In contrast, models such as *o1* show the ability to solve such problems, though the underlying mechanisms remain unclear.

On the other hand, LLM-based agents have demonstrated remarkable competence in performing real-world planning tasks, even without RL training (Wang et al., 2024a). Many of these planning tasks can be naturally abstracted as path planning problems on a graph. For example, in tool-augmented agents (Shen et al., 2023), tool dependencies can be modeled as a graph where nodes represent tools and edges represent dependency relations (Wu et al., 2024b). Planning, in this context, involves finding a path of tools to fulfill the user's request. Similarly, in mathematical reasoning agents (Trinh et al., 2024), theorem dependencies form a graph where constructing a proof is equivalent to finding a path. In game-playing agents such as Voyager (Wang et al., 2023a), skill dependencies create a graph structure where planning determines the sequence of skills needed to accomplish tasks. These observations motivate our abstraction of planning as a path planning problem in this work.

Agents trained without RL face two key challenges: (1) supervised fine-tuning loss is misaligned with the agent's ultimate objectives, and (2) real-world data is scarce. RL addresses the first issue by explicitly optimizing for the end goal through a reward signal, and the second by generating exploratory data. Consequently, RL significantly mitigates these limitations and improves performance (Zhang et al., 2025a). Our paper further examines the benefits of RL over SFT, as well as the limitations of RL, providing insights for future research directions.

### B.2    RL FOR LLMS

Recently, RL has been widely adopted to enhance reasoning capabilities in language models, exemplified by milestone systems such as OpenAI's *o1* and DeepSeek-R1. This paradigm has inspired a new wave of reasoning-focused models, including Qwen-3 and Phi-4 Reasoning (Zhang et al., 2025b). State-of-the-art LLM-based agents also commonly employ RL (Zhang et al., 2025a).

Despite its empirical success, the mechanisms by which RL improves LLM performance remain an active area of research, with current understanding scattered across multiple works. For instance, Chu et al. (2025) empirically compares SFT and RL on reasoning benchmarks, concluding that RL provides better generalization. Theoretical analysis in (Setlur et al., 2025) further shows that any verification-free approach, such as SFT, is suboptimal. Additionally, Yue et al. (2025) identifies an *entropy mechanism*, establishing and empirically validating a trade-off between entropy and accuracy during RL training.

In this paper, we focus on path planning as a case study and derive results consistent with prior work: (1) SFT tends to memorize training data and produce co-occurrence-driven outputs; (2) RL surpasses SFT primarily through exploration; and (3) diversity collapse occurs during PG training. Beyond these findings, we uncover evidence suggesting that Q-learning may offer advantages over policy gradient methods, introducing a new perspective on RL for LLMs.

### B.3 GRAPH PROBLEMS WITH LANGUAGE MODELS

Graph problems serve as a valuable testbed for analyzing the reasoning capabilities of language models. From an empirical standpoint, several benchmarks have been proposed (Guo et al., 2023; Wang et al., 2023b; Dai et al., 2024), spanning a spectrum of tasks: classic graph problems (e.g., connectivity, path-finding, and pattern detection), graph neural network (GNN) benchmarks (e.g., node and graph classification), and semantic graph-based question answering (e.g., on knowledge graphs). Without additional training, LLMs generally underperform on these tasks. To improve performance, early approaches leverage instruction tuning and DPO (Luo et al., 2024; Chen et al., 2024a; Perozzi et al., 2024; Chai et al., 2023; Chen et al., 2024b), while later methods employ RL (Guo et al., 2025), which consistently achieves superior results.

There are three major paradigms for analyzing how transformers solve graph-related reasoning tasks. The first is *mechanistic interpretability*, which reverse-engineers trained model weights (Neel et al., 2023). For example, Cohen et al. (2025) observed that transformers implement a spectral algorithm to compute shortest paths. However, this paradigm largely relies on empirical observation without theoretical grounding. The second paradigm is based on *expressiveness* analysis (Dai et al., 2024; Sanford et al., 2024; Dai et al., 2025; De Luca & Fountoulakis, 2024), constructing weight configurations that enable transformers to simulate algorithms. Yet, such configurations are often unrealistic for transformers trained via SGD (e.g., embedding vectors explicitly set to $1, 2, \ldots, L$ (Dai et al., 2024)). The third paradigm investigates *gradient dynamics*, which is both practical and challenging due to the non-convexity of the optimization landscape. Prior work has analyzed path-finding in directed graphs (Wang et al., 2024b) and compositionality of paths (Zhu et al., 2024).

To the best of our knowledge, this work presents the first analysis of RL gradient dynamics in LLMs. Our results explain why RL-based methods outperform SFT approaches and highlight the potential advantages of Q-learning–driven methods, opening promising directions for future research.

## C   APPENDIX FOR SFT

### C.1   PATH PLANNING ALGORITHM IN TRANSFORMER

---

**Algorithm 1** A handcrafted path planning algorithm

---

1: **Input:** Adjacency matrix $\boldsymbol{A}$, reachability matrix $\boldsymbol{R}$, source node $s$, target node $t$
2: Set path $P = [s\,t\,s]$ and set current node $i = s$
3: **while** $i \neq t$ **do**
4:     Obtain $S = \{k | \boldsymbol{A}_{(i,k)} = 1 \text{ and } \boldsymbol{R}_{(t,k)} = 1\}$
5:     Randomly sample next node $k$ from $S$
6:     Append $k$ to path $P$, and set $i = k$
7: **end while**
8: **output** path $P$

---

### C.2   PROOF OF THEOREM 3.1

*Proof.* The next-token prediction cross-entropy loss can be written as

$$\ell = -\sum_{\mathbf{u} \in \mathcal{D}^{\mathrm{SFT}}} \sum_{m \geq 1} \sum_k \delta_{k=u_{m+1}} \log \hat{\mathbf{u}}_m[k].$$

Under the assumption that the output distribution $\hat{\mathbf{u}}_m$ depends only on the target node $u_{\mathrm{target}}$ and the current node $u_m$, we can aggregate identical terms, and express the loss as

$$-\sum_{u_{\mathrm{target}}, u_m} \left( \sum_{k'} N_{u_{\mathrm{target}}, u_m, k'} \right) \left( \sum_k \frac{N_{u_{\mathrm{target}}, u_m, k}}{\sum_{k'} N_{u_{\mathrm{target}}, u_m, k'}} \log \frac{\exp(\mathbf{f}(u_{\mathrm{target}}, u_m)[k])}{\sum_{k'} \exp(\mathbf{f}(u_{\mathrm{target}}, u_m)[k'])} \right).$$

If $\sum_{k'} N_{u_{\text{target}}, u_m, k'} \neq 0$, the expression in brackets is the cross-entropy between the empirical distribution and the distribution of doing softmax on vector $\mathbf{f}(u_{\text{target}}, u_m)$. It is minimized when

$$\frac{\exp(\mathbf{f}(u_{\text{target}}, u_m)[k])}{\sum_{k'} \exp(\mathbf{f}(u_{\text{target}}, u_m)[k'])} = \frac{N_{u_{\text{target}}, u_m, k}}{\sum_{k'} N_{u_{\text{target}}, u_m, k'}}.$$

If $\sum_{k'} N_{u_{\text{target}}, u_m, k'} = 0$, the loss does not depend on $\mathbf{f}(u_{\text{target}}, u_m)$, so it can be any valid probability distribution. $\square$

## D  APPENDIX FOR POLICY GRADIENT

### D.1  PROOF OF THEOREM 4.1

*Proof.* In this case, the loss function of policy gradient is

$$\ell = \sum_{\mathbf{u} \in \mathcal{D}^{\text{RL,t}}} \delta_{\mathbf{u} \in \mathcal{P}} \left( -\sum_{m \geq 1} \log \hat{\mathbf{u}}_m[u_{m+1}] \right) = \sum_{\mathbf{u} \in \mathcal{D}^{\text{RL,t}} \cap \mathcal{P}} \left( -\sum_{m \geq 1} \log \hat{\mathbf{u}}_m[u_{m+1}] \right).$$

$\square$

### D.2  PROOF OF THEOREM 4.2

*Proof.* We first rewrite the loss function as

$$\ell = \sum_{\mathbf{u} \in \mathcal{D}^{\text{RL,t}} \cap \mathcal{P}} \left( -\sum_{m \geq 1} \log \hat{\mathbf{u}}_m[u_{m+1}] \right) = \sum_{i,j,k} N_{i,j,k}^{R,P,t} \left( -\log \frac{\exp(\mathbf{f}(i,j)[k])}{\sum_{k'} \exp(\mathbf{f}(i,j)[k'])} \right),$$

where $N_{i,j,k}^{R,P,t}$ denote the number of times that $u_{\text{target}} = i$, $u_m = j$ and $u_{m+1} = k$ for $m \geq 1$ in set $\mathcal{D}^{\text{RL,t}} \cap \mathcal{P}$. Then we can take the gradient and get

$$\frac{\partial \ell}{\partial \mathbf{f}(i,j)[k]} = -N_{i,j,k}^{R,P,t} + \frac{\exp(\mathbf{f}(i,j)[k])}{\sum_{k'} \exp(\mathbf{f}(i,j)[k'])} \sum_{k'} N_{i,j,k'}^{R,P,t}.$$

For a wrong tuple $i, j, k$ (where $k$ is not adjacent with $j$ or $k$ cannot reach $i$), $N_{i,j,k}^{R,P,t}$ is always zero. Thus, the gradient is always positive. On the other hand, we will also have

$$\sum_k \frac{\partial \ell}{\partial \mathbf{f}(i,j)[k]} = -\sum_k N_{i,j,k}^{R,P,t} + \frac{\sum_k \exp(\mathbf{f}(i,j)[k])}{\sum_{k'} \exp(\mathbf{f}(i,j)[k'])} \sum_{k'} N_{i,j,k'}^{R,P,t} = 0.$$

$\square$

### D.3  PROOF OF THEOREM 4.3

*Proof.* Note that

$$
\begin{aligned}
KL(U_{C(i,j)} \| \mathbf{softmax}(\mathbf{f}(i,j))) &= \sum_{k \in C(i,j)} \frac{1}{|C(i,j)|} \left( -\log |C(i,j)| - \log \frac{\exp(\mathbf{f}(i,j)[k])}{\sum_{k' \in C(i,j)} \exp(\mathbf{f}(i,j)[k'])} \right) \\
&= -\log |C(i,j)| - \frac{1}{|C(i,j)|} \sum_{k \in C(i,j)} \log \frac{\exp(\mathbf{f}(i,j)[k])}{\sum_{k' \in C(i,j)} \exp(\mathbf{f}(i,j)[k'])}.
\end{aligned}
$$

Thus, it is sufficient to prove

$$\sum_{k \in C(i,j)} \log \frac{\exp(\mathbf{f}^t(i,j)[k])}{\sum_{k' \in C(i,j)} \exp(\mathbf{f}^t(i,j)[k'])} \geq \mathbb{E} \left[ \sum_{k \in C(i,j)} \log \frac{\exp(\mathbf{f}^{t+1}(i,j)[k])}{\sum_{k' \in C(i,j)} \exp(\mathbf{f}^{t+1}(i,j)[k'])} \right].$$

According to the gradient, we have that

$$\mathbf{f}^{t+1}(i,j)[k] = \mathbf{f}^t(i,j)[k] - \eta\left(-N_{i,j,k}^{R,P,t} + \frac{\exp(\mathbf{f}(i,j)[k])}{\sum_{k'\in C(i,j)}\exp(\mathbf{f}(i,j)[k'])}\sum_{k'\in C(i,j)}N_{i,j,k'}^{R,P,t}\right),$$

Here $\eta$ is the step size.

Let $N_{i,j}^{R,P,t} = \sum_{k'\in C(i,j)}N_{i,j,k'}^{R,P,t}$, then due to the on-policy updating in policy gradient, we know that $N_{i,j,k}^{R,P,t}$ is the counter of outcome $k$ for $N_{i,j}^{R,P,t}$ independent multi-nomial random variables with parameters $\left\{\frac{\exp(\mathbf{f}^t(i,j)[k])}{\sum_{k'\in C(i,j)}\exp(\mathbf{f}^t(i,j)[k'])}\right\}_{k\in C(i,j)}$. This means that

$$\mathbb{E}[\mathbf{f}^{t+1}(i,j)[k]] = \mathbf{f}^t(i,j)[k].$$

Moreover, since $\log\left(\sum_{k'\in C(i,j)}\exp(\mathbf{f}^t(i,j)[k'])\right)$ is a convex function, we also have

$$\mathbb{E}\left[\log\left(\sum_{k'\in C(i,j)}\exp(\mathbf{f}^{t+1}(i,j)[k'])\right)\right] \geq \log\left(\sum_{k'\in C(i,j)}\exp(\mathbb{E}[\mathbf{f}^{t+1}(i,j)[k']])\right)$$
$$= \log\left(\sum_{k'\in C(i,j)}\exp(\mathbf{f}^t(i,j)[k'])\right)$$

Because of this, we have

$$\sum_{k\in C(i,j)}\log\frac{\exp(\mathbf{f}^t(i,j)[k])}{\sum_{k'\in C(i,j)}\exp(\mathbf{f}^t(i,j)[k'])} - \mathbb{E}\left[\sum_{k\in C(i,j)}\log\frac{\exp(\mathbf{f}^{t+1}(i,j)[k])}{\sum_{k'\in C(i,j)}\exp(\mathbf{f}^{t+1}(i,j)[k'])}\right]$$

$$= \sum_{k\in C(i,j)}\mathbf{f}^t(i,j)[k] - \mathbb{E}\left[\sum_{k\in C(i,j)}\mathbf{f}^{t+1}(i,j)[k]\right] - |C(i,j)|\log\left(\sum_{k'\in C(i,j)}\exp(\mathbf{f}^t(i,j)[k'])\right)$$

$$+ |C(i,j)|\mathbb{E}\left[\log\left(\sum_{k'\in C(i,j)}\exp(\mathbf{f}^{t+1}(i,j)[k'])\right)\right]$$

$$\geq 0.$$

$\square$

## D.4 Proof of Theorem 4.4

*Proof.* When $\lambda > 0$, the loss function is

$$\ell = \sum_{i,j,k}N_{i,j,k}^{R,P,t}\left(-\log\mathbf{q}(i,j)[k]\right) + \lambda\sum_{i,j,k}N_{i,j,k}^{R,t}\left(\log\mathbf{q}(i,j)[k]\left\{\log\frac{\mathbf{q}(i,j)[k]}{\mathbf{q}^{\text{base}}(i,j)[k]}\right\}\right),$$

where $N_{i,j,k}^{R,t}$ denote the number of times that $u_{\text{target}} = i$, $u_m = j$ and $u_{m+1} = k$ for $m \geq 1$ in set $\mathcal{D}^{\text{RL,t}}$.

We can take the gradient and get:

$$\frac{\partial\ell}{\partial\mathbf{f}(i,j)[k]} = -N_{i,j,k}^{R,P,t} + \mathbf{q}(i,j)[k]\sum_{k'}N_{i,j,k'}^{R,P,t} + \lambda N_{i,j,k}^{R,t}(1-\mathbf{q}(i,j)[k])\log\frac{\mathbf{q}(i,j)[k]}{\mathbf{q}^{\text{base}}(i,j)[k]}$$

$$- \lambda\sum_{k'\neq k}N_{i,j,k'}^{R,t}\mathbf{q}(i,j)[k]\log\frac{\mathbf{q}(i,j)[k]}{\mathbf{q}^{\text{base}}(i,j)[k]}.$$

Taking expectation, we can get:

$$
\begin{aligned}
\mathbb{E}\left[\frac{\partial \ell}{\partial \mathbf{f}(i,j)[k]}\right] &= -\mathbb{E}[N_{i,j,k}^{R,P,t}] + \mathbf{q}(i,j)[k]\sum_{k'}\mathbb{E}[N_{i,j,k'}^{R,P,t}] + \lambda\mathbb{E}[N_{i,j,k}^{R,t}](1-\mathbf{q}(i,j)[k])\log\frac{\mathbf{q}(i,j)[k]}{\mathbf{q}^{\text{base}}(i,j)[k]} \\
&\quad -\lambda\sum_{k'\neq k}\mathbb{E}[N_{i,j,k'}^{R,t}]\mathbf{q}(i,j)[k]\log\frac{\mathbf{q}(i,j)[k]}{\mathbf{q}^{\text{base}}(i,j)[k]}.
\end{aligned}
$$

Letting $N_{i,j}^{R,t} = \sum_k N_{i,j,k}^{R,t}$, then due to on-policy training, we have that $\mathbb{E}[N_{i,j,k}^{R,P,t}] = N_{i,j}^{R,t}\mathbf{q}(i,j)[k]\mathbf{p}(i,j)[k]$, and $\mathbb{E}[N_{i,j,k}^{R,t}] = N_{i,j}^{R,t}\mathbf{q}(i,j)[k]$.

Hence

$$
\begin{aligned}
&\mathbb{E}\left[\frac{\partial \ell}{\partial \mathbf{f}(i,j)[k]}\right] \\
&= -N_{i,j}^{R,t}\mathbf{q}(i,j)[k]\mathbf{p}(i,j)[k] + N_{i,j}^{R,t}\mathbf{q}(i,j)[k]\sum_{k'}\mathbf{q}(i,j)[k']\mathbf{p}(i,j)[k'] \\
&\quad +\lambda N_{i,j}^{R,t}\mathbf{q}(i,j)[k](1-\mathbf{q}(i,j)[k])\log\frac{\mathbf{q}(i,j)[k]}{\mathbf{q}^{\text{base}}(i,j)[k]} - \lambda N_{i,j}^{R,t}\sum_{k'\neq k}\mathbf{q}(i,j)[k]\mathbf{q}(i,j)[k']\log\frac{\mathbf{q}(i,j)[k]}{\mathbf{q}^{\text{base}}(i,j)[k]} \\
&= N_{i,j}^{R,t}\mathbf{q}(i,j)[k]\sum_{k'}\mathbf{q}(i,j)[k']\left(\mathbf{p}(i,j)[k]-\mathbf{p}(i,j)[k']+\lambda\log\frac{\mathbf{q}(i,j)[k]}{\mathbf{q}^{\text{base}}(i,j)[k]}-\lambda\log\frac{\mathbf{q}(i,j)[k']}{\mathbf{q}^{\text{base}}(i,j)[k']}\right).
\end{aligned}
$$

The stable point must satisfy that, for any tuple $i,j,k$, $\mathbb{E}\left[\frac{\partial \ell}{\partial \mathbf{f}(i,j)[k]}\right] = 0$. And we claim that in this case, for fixed $i,j$ and any $k'$ such that $\mathbf{q}(i,j)[k'] > 0$, their $\mathbf{p}(i,j)[k'] + \lambda\log\frac{\mathbf{q}(i,j)[k']}{\mathbf{q}^{\text{base}}(i,j)[k']}$ should equal. Otherwise we can always look for $k^* = \arg\min_{k':\mathbf{q}(i,j)[k']>0}\mathbf{p}(i,j)[k'] + \lambda\log\frac{\mathbf{q}(i,j)[k']}{\mathbf{q}^{\text{base}}(i,j)[k']}$, and its expected gradient $\mathbb{E}\left[\frac{\partial \ell}{\partial \mathbf{f}(i,j)[k^*]}\right]$ is strict negative. $\square$

# E  APPENDIX FOR Q-LEARNING

## E.1  PROOF OF LEMMA 5.1

*Proof.* Fix any triple $(i,j,k)$. Consider training sequences whose first two nodes satisfy $u_{\text{source}} \in \mathcal{V}$ and $u_{\text{target}} = i$. By the definition of the training process, $\mathbb{P}(u_{\text{source}} \in \mathcal{V}, u_{\text{target}} = i) > 0$. Under $\epsilon$-exploration uniform over $\mathcal{V}$,

$$\forall v \in \mathcal{V}: \quad \mathbb{P}(\text{next node} = v) \geq \epsilon/|\mathcal{V}|.$$

Condition on the event $\{u_{\text{source}} \in \mathcal{V}, u_{\text{target}} = i\}$. Then in the next two decisions,

$$\mathbb{P}(u_2 = j \mid u_{\text{source}} \in \mathcal{V}, u_{\text{target}} = i) \geq \epsilon/|\mathcal{V}|, \quad \mathbb{P}(u_3 = k \mid u_{\text{source}} \in \mathcal{V}, u_{\text{target}} = i, u_2 = j) \geq \epsilon/|\mathcal{V}|.$$

Hence

$$\mathbb{P}(u_{\text{target}} = i, u_2 = j, u_3 = k) \geq p_0(\epsilon/|\mathcal{V}|)^2 > 0.$$

Each occurrence of $(u_{\text{target}}, u_2, u_3) = (i,j,k)$ triggers one update of $\mathbf{f}(i,j)[k]$. Since each occurrence yields an update of $\mathbf{f}(i,j)[k]$, we obtain

$$\liminf_{T\to\infty}\frac{1}{T}\sum_{t=0}^{T-1}\delta_{(i_t,j_t,k_t)=(i,j,k)} \geq \underline{N}_{i,j,k}^{\text{prop}} > 0,$$

which is the persistent exploration condition. $\square$

## E.2  PROOF OF THEOREM 5.1

*Proof.* At a stable point, the expected update of each coordinate vanishes. The per-step loss is

$$\ell = \left(\mathbf{f}(u_{\text{target}}, u_m)[u_{m+1}] - \delta_{\mathbf{u}\in\mathcal{P}}\,\delta_{u_{m+1}=u_{\text{target}}} - \{\max_{k'}\mathbf{f}(u_{\text{target}}, u_{m+1})[k']\}\right)^2.$$

Taking the gradient with respect to $\mathbf{f}(u_{\text{target}}, u_m)[u_{m+1}]$ and setting the expectation to zero yields, for every triple $(i, j, k)$,

$$\mathbf{f}(i, j)[k] = \mathbb{E}[\delta_{\mathbf{u} \in \mathcal{P}} \, \delta_{k=i}] + \max_{k'} \mathbf{f}(i, k)[k']. \tag{4}$$

If $k \neq i$, the expectation term in equation 4 vanishes, so

$$\mathbf{f}(i, j)[k] = \max_{k'} \mathbf{f}(i, k)[k'],$$

which does not depend on $j$. Thus, for each $k \neq i$, we have

$$\mathbf{f}(i, j)[k] = \max_{k'} \mathbf{f}(i, k)[k'] = \max_{k' \neq i} \left( \max_{k''} \mathbf{f}(i, k')[k''], \, \mathbf{f}(i, k)[i] \right) = \max_{k' \neq i} \max_{k''} \mathbf{f}(i, k')[k''].$$

Here the second equality separates the case $k' = i$ from $k' \neq i$: if $k' = i$, the chain terminates immediately, corresponding to the $\cdots \to k \to i$ path, so we take the value $\mathbf{f}(i, k)[i]$ directly. If $k' \neq i$, we must expand one step further, leading to $\max_{k''} \mathbf{f}(i, k')[k'']$. The last equality then observes that including the $k' = i$ term does not change the maximization, since it is already dominated by the expansion over $k' \neq i$. Thus the final expression simply enumerates all possible terms with the restriction $k' \neq i$. This expression no longer depends on $k$ or $j$.

Therefore, for each fixed $i$ and $k \neq i$, all $\mathbf{f}(i, j)[k]$ take a common value $c_i$, independent of $j$ and $k$. $\qquad\square$

### E.3 PROOF OF THEOREM 5.2

*Proof.* For clarity, we introduce two notations that will be used repeatedly. First, for $i, k \in [n]$ and iteration $t$, define

$$S_{i,k}^{(t)} := \max_{k'} \mathbf{f}^{(t)}(i, k)[k'].$$

Second, we write $k \in \text{Anc}(i)$ if there exists $m \geq 1$ such that $(\mathbf{A}^m)[k, i] = 1$, i.e. $k$ is an ancestor of $i$.

The per-step loss under the process reward is

$$\ell = \left( \mathbf{f}(u_{\text{target}}, u_m)[u_{m+1}] - (\delta_{u_{m+1}=u_{\text{target}}} - \delta_{(u_m, u_{m+1}) \notin \mathcal{E}}) - \{ \max_{k'} \mathbf{f}(u_{\text{target}}, u_{m+1})[k'] \} \right)^2.$$

Taking the gradient with respect to the active coordinate $\mathbf{f}(u_{\text{target}}, u_m)[u_{m+1}]$ gives

$$\frac{\partial \ell}{\partial \mathbf{f}(u_{\text{target}}, u_m)[u_{m+1}]} = 2 \left( \mathbf{f}(u_{\text{target}}, u_m)[u_{m+1}] - \delta_{u_{m+1}=u_{\text{target}}} + \delta_{(u_m, u_{m+1}) \notin \mathcal{E}} - \max_{k'} \mathbf{f}(u_{\text{target}}, u_{m+1})[k'] \right).$$

Applying gradient descent with learning rate $\eta$ yields

$$\mathbf{f}(u_{\text{target}}, u_m)[u_{m+1}] \leftarrow (1 - 2\eta) \, \mathbf{f}(u_{\text{target}}, u_m)[u_{m+1}] + 2\eta \left( \delta_{u_{m+1}=u_{\text{target}}} - \delta_{(u_m, u_{m+1}) \notin \mathcal{E}} + \max_{k'} \mathbf{f}(u_{\text{target}}, u_{m+1})[k'] \right).$$

Renaming $(i, j, k) = (u_{\text{target}}, u_m, u_{m+1})$ and writing $S_{i,k}^{(t)} = \max_{k'} \mathbf{f}^{(t)}(i, k)[k']$, the recursion is

$$\mathbf{f}^{(t+1)}(i, j)[k] = (1 - 2\eta)\mathbf{f}^{(t)}(i, j)[k] + 2\eta(\delta_{k=i} + (\mathbf{A}[j, k] - 1) + S_{i,k}^{(t)}). \tag{$\star$}$$

When $k = i$, by convention $S_{i,i}^{(t)} = 0$, so the update is

$$\mathbf{f}^{(t+1)}(i, j)[i] = (1 - 2\eta)\mathbf{f}^{(t)}(i, j)[i] + 2\eta \, \mathbf{A}[j, i].$$

This linear recursion has fixed point $\mathbf{A}[j, i]$ and solution

$$\mathbf{f}^{(t)}(i, j)[i] = (1 - (1 - 2\eta)^t) \, \mathbf{A}[j, i],$$

which converges to 1 if $\mathbf{A}[j, i] = 1$ and to 0 otherwise. The contraction factor is $|1 - 2\eta|$.

For $k \neq i$, the limit of $S_{i,k}^{(t)}$ must be analyzed. If $k \in \text{Anc}(i) \setminus \{i\}$, then either $k$ is a parent of $i$, in which case $\mathbf{f}^{(t)}(i, k)[i] \to 1$ and hence $S_{i,k}^{(t)} \to 1$, or $k$ has a child $r$ with $r \in \text{Anc}(i)$.

Inductively $S_{i,r}^{(t)} \to 1$, and then $(\star)$ implies $\mathbf{f}^{(t)}(i,k)[r] \to 1$, so $S_{i,k}^{(t)} \to 1$. If $k \notin \mathrm{Anc}(i)$, then all children $r$ of $k$ also satisfy $r \notin \mathrm{Anc}(i)$, and inductively $S_{i,r}^{(t)} \to 0$, giving $\mathbf{f}^{(t+1)}(i,k)[r] = (1-2\eta)\mathbf{f}^{(t)}(i,k)[r]+2\eta S_{i,r}^{(t)} \to 0$. Thus $S_{i,k}^{(t)} \to 0$. Therefore the limit is $S_{i,k}^{(t)} \to 1$ if $k \in \mathrm{Anc}(i)\backslash\{i\}$ and $S_{i,k}^{(t)} \to 0$ otherwise.

For $k \neq i$, substituting the limiting $S_{i,k}^{(t)}$ into $(\star)$ gives
$$\mathbf{f}^{(t+1)}(i,j)[k] = (1-2\eta)\mathbf{f}^{(t)}(i,j)[k] + 2\eta((\mathbf{A}[j,k]-1) + S_{i,k}^{(t)}).$$

If $\mathbf{A}[j,k]=1$ and $k \in \mathrm{Anc}(i)$, then $S_{i,k}^{(t)} \to 1$, so $\mathbf{f}^{(t)}(i,j)[k] \to 1$. If $\mathbf{A}[j,k]=1$ and $k \notin \mathrm{Anc}(i)$, then $S_{i,k}^{(t)} \to 0$, so $\mathbf{f}^{(t)}(i,j)[k] \to 0$. If $\mathbf{A}[j,k]=0$ and $k \in \mathrm{Anc}(i)$, then $S_{i,k}^{(t)} \to 1$, so the recursion is $\mathbf{f}^{(t+1)}(i,j)[k] = (1-2\eta)\mathbf{f}^{(t)}(i,j)[k]$, implying $\mathbf{f}^{(t)}(i,j)[k] \to 0$. If $\mathbf{A}[j,k]=0$ and $k \notin \mathrm{Anc}(i)$, then $S_{i,k}^{(t)} \to 0$, so the recursion is $\mathbf{f}^{(t+1)}(i,j)[k] = (1-2\eta)\mathbf{f}^{(t)}(i,j)[k] - 2\eta$, which converges to $-1$. These limits match the cases in the theorem.

To establish rates, define the weight error $e_t^W(i,j,k) = \mathbf{f}^{(t)}(i,j)[k] - \mathbf{f}^\star(i,j)[k]$ and the max error $e_t^S(i,k) = S_{i,k}^{(t)} - S_{i,k}^\star$. When $k = i$, the recursion is
$$e_{t+1}^W(i,j,i) = (1-2\eta)\, e_t^W(i,j,i),$$
so each update contracts the error by $|1-2\eta|$. Under persistent exploration, the coordinate $(i,j,i)$ is updated with positive frequency $\underline{N}_{i,j,i}^{\mathrm{prop}}$, so in global time
$$|e_t^W(i,j,i)| \leq C\,(|1-2\eta|^{\underline{N}_{i,j,i}^{\mathrm{prop}}-\varepsilon})^t$$
for any $\varepsilon > 0$ and large enough $t$.

When $k \neq i$, the recursion is
$$e_{t+1}^W(i,j,k) = (1-2\eta)\, e_t^W(i,j,k) + 2\eta\, e_t^S(i,k).$$
The error $e_t^S(i,k)$ depends only on $\{e_t^W(i,k,r)\ :\ r \text{ is a child of } k\}$. Along any directed path $k = v_0 \to v_1 \to \cdots \to v_m = i$, the error at $k$ can decay only after the error at $v_1$ has already decayed, and so on. Thus, the effective contraction factor for $e_t^W(i,j,k)$ is the product of the per-edge contraction rates
$$\prod_{n=0}^{m-1}(|1-2\eta|^{\underline{N}_{i,v_n,v_{n+1}}^{\mathrm{prop}}}).$$
Formally, by induction, for any $\varepsilon > 0$ and sufficiently large $t$ we have
$$|e_t^W(i,j,k)| \leq C\left(\max_{\text{paths } p:k\to i}\ \prod_{(v_n,v_{n+1})\in p}|1-2\eta|^{\underline{N}_{i,v_n,v_{n+1}}^{\mathrm{prop}}-\varepsilon}\right)^t.$$

Therefore, all iterates converge linearly in global time, with effective rates determined jointly by $\eta$ and the update proportions $\underline{N}_{i,j,k}^{\mathrm{prop}}$. This completes the proof. $\qquad\square$

### E.4 PROOF OF THEOREM 5.3

*Proof.* Let $N_{i,j,k}^{\mathrm{prop}}$ denote the asymptotic proportion of triples $(u_{\text{target}}, u_m, u_{m+1}) = (i,j,k)$ occurring in the generated sequences at the stable point, under the given sampling method and the persistent exploration condition. Equivalently, $N_{i,j,k}^{\mathrm{prop}}$ is the limiting frequency with which state $j$ transitions to $k$ with target $i$ in the trajectories sampled by the model. By definition $N_{i,j,k}^{\mathrm{prop}} > 0$ for all $i,j,k$.

At a stable point of the updates, the expected gradient with respect to each parameter must vanish. Averaging the stationarity conditions with weights $N_{i,j,k}^{\mathrm{prop}}$ yields, for all $i,j$,
$$\sum_j N_{i,j,k}^{\mathrm{prop}}\Big(\mathbf{W}^M[j,k] + \mathbf{W}^V[i,k] - \mathbf{A}[j,k] + 1 - \delta_{i=k} - \max_{k'}(\mathbf{W}^M[k,k'] + \mathbf{W}^V[i,k'])\Big) = 0,$$
$$\sum_i N_{i,j,k}^{\mathrm{prop}}\Big(\mathbf{W}^M[j,k] + \mathbf{W}^V[i,k] - \mathbf{A}[j,k] + 1 - \delta_{i=k} - \max_{k'}(\mathbf{W}^M[k,k'] + \mathbf{W}^V[i,k'])\Big) = 0.$$

Introduce centered variables
$$\mathbf{S}_{j,k} := \mathbf{W}^M[j,k] - \mathbf{A}[j,k] + 1, \qquad \mathbf{T}_{i,k} := \mathbf{W}^V[i,k] - \delta_{i=k} - \max_{k'}(\mathbf{W}^M[k,k'] + \mathbf{W}^V[i,k']),$$

so that normalizing each sum by its positive denominator gives the block system
$$\mathbf{S}_k + \mathbf{P}_k\,\mathbf{T}_k = \mathbf{0}, \qquad \mathbf{T}_k + \mathbf{Q}_k\,\mathbf{S}_k = \mathbf{0}, \tag{5}$$
where $\mathbf{S}_k = (\mathbf{S}_{j,k})_{j\in[n]}$, $\mathbf{T}_k = (\mathbf{T}_{i,k})_{i\in[n]}$, and
$$(\mathbf{P}_k)[j,i] = \frac{N^{\mathrm{prop}}_{i,j,k}}{\sum_{i'} N^{\mathrm{prop}}_{i',j,k}}, \qquad (\mathbf{Q}_k)[i,j] = \frac{N^{\mathrm{prop}}_{i,j,k}}{\sum_{j'} N^{\mathrm{prop}}_{i,j',k}}.$$

Since $N^{\mathrm{prop}}_{i,j,k} > 0$, every entry of $\mathbf{P}_k, \mathbf{Q}_k$ is strictly positive, and both are row-stochastic. Hence $\mathbf{P}_k\mathbf{Q}_k$ and $\mathbf{Q}_k\mathbf{P}_k$ are strictly positive stochastic matrices. By the Perron–Frobenius theorem, both have a simple eigenvalue $1$ with eigenvector $\mathbf{1}$, and all other eigenvalues satisfy $|\lambda| < 1$. Thus
$$\ker(\mathbf{I} - \mathbf{P}_k\mathbf{Q}_k) = \mathrm{span}\{\mathbf{1}\}, \qquad \ker(\mathbf{I} - \mathbf{Q}_k\mathbf{P}_k) = \mathrm{span}\{\mathbf{1}\}.$$
From equation 5, eliminating $\mathbf{T}_k$ yields $\mathbf{S}_k = (\mathbf{P}_k\mathbf{Q}_k)\mathbf{S}_k$, so $\mathbf{S}_k = c_k\mathbf{1}$ for some $c_k \in \mathbb{R}$, and then $\mathbf{T}_k = -\mathbf{Q}_k\mathbf{S}_k = -c_k\mathbf{1}$. Returning to the definitions,
$$\mathbf{W}^M[j,k] = \mathbf{A}[j,k] - 1 + c_k, \qquad \mathbf{W}^V[i,k] = \delta_{i=k} + \max_{k'}(\mathbf{W}^M[k,k'] + \mathbf{W}^V[i,k']) - c_k.$$

Substituting $\mathbf{W}^M[k,k'] = \mathbf{A}[k,k'] - 1 + c_k$ and writing $\mathbf{V}_{i,k'} := \mathbf{W}^V[i,k'] - c_k$ gives
$$\mathbf{V}_{i,k} = \delta_{i=k} + \max_{k':\,\mathbf{A}[k,k']=1} \mathbf{V}_{i,k'}.$$

On a DAG, the unique $\{0,1\}$ solution of this recursion is the reachability indicator $\mathbf{R}_{i,k}$. An induction over a topological order shows $\mathbf{V}_{i,k} = \mathbf{R}_{i,k}$ for all $i,k$. Therefore
$$\mathbf{W}^V[i,k] = \mathbf{R}[i,k] - c_k.$$

Finally, note that if $(\mathbf{S}_k, \mathbf{T}_k)$ solves equation 5, then so does $(\mathbf{S}_k + c\mathbf{1}, \mathbf{T}_k - c\mathbf{1})$ for any $c \in \mathbb{R}$, since $\mathbf{P}_k\mathbf{1} = \mathbf{Q}_k\mathbf{1} = \mathbf{1}$. Hence, the solution set for each $k$ is exactly a one-dimensional affine line parametrized by $c_k$.

Conversely, if $(\mathbf{W}^M, \mathbf{W}^V)$ is of the above form, then plugging it into the update equations shows that the expected increment is identically zero: both sides of the gradient equations cancel by construction, so the point is stationary. Therefore, these conditions are not only necessary but also sufficient for stability. $\qquad\square$

## F   EQUIVALENCE OF UNCLIPPED PPO AND POLICY GRADIENT

For a sequence $\mathbf{u}$, the policy gradient objective is
$$\ell_{\mathrm{PG}}(\mathbf{u}) = -\sum_{m\geq 1} R(\mathbf{u})\log \hat{\mathbf{u}}_m[u_{m+1}],$$
where $R(\mathbf{u}) = r\,\delta_{\mathbf{u}\in\mathcal{P}} + p$. Taking the gradient gives
$$\nabla_\theta \ell_{\mathrm{PG}}(\mathbf{u}) = -\sum_{m\geq 1} R(\mathbf{u})\nabla_\theta \log \hat{\mathbf{u}}_m[u_{m+1}]$$
$$= -\sum_{m\geq 1} R(\mathbf{u})\frac{\nabla_\theta \hat{\mathbf{u}}_m[u_{m+1}]}{\hat{\mathbf{u}}_m[u_{m+1}]}.$$

For unclipped PPO, the ratio between new and old probabilities is formed, with the denominator detached. The loss is
$$\ell_{\mathrm{PPO\text{-}uc}}(\mathbf{u}) = -\sum_{m\geq 1} R(\mathbf{u})\frac{\hat{\mathbf{u}}_m[u_{m+1}]}{\{\hat{\mathbf{u}}_m[u_{m+1}]\}}.$$

Since the denominator $\{\hat{\mathbf{u}}_m[u_{m+1}]\}$ is treated as constant, its gradient vanishes. Thus
$$\nabla_\theta \ell_{\mathrm{PPO\text{-}uc}}(\mathbf{u}) = -\sum_{m\geq 1} R(\mathbf{u})\frac{\nabla_\theta \hat{\mathbf{u}}_m[u_{m+1}]}{\{\hat{\mathbf{u}}_m[u_{m+1}]\}}.$$

Comparing with the policy gradient expression, we see the two gradients coincide. Therefore, for any fixed sequence $\mathbf{u}$, unclipped PPO with a stop-gradient denominator is exactly equivalent to vanilla policy gradient.

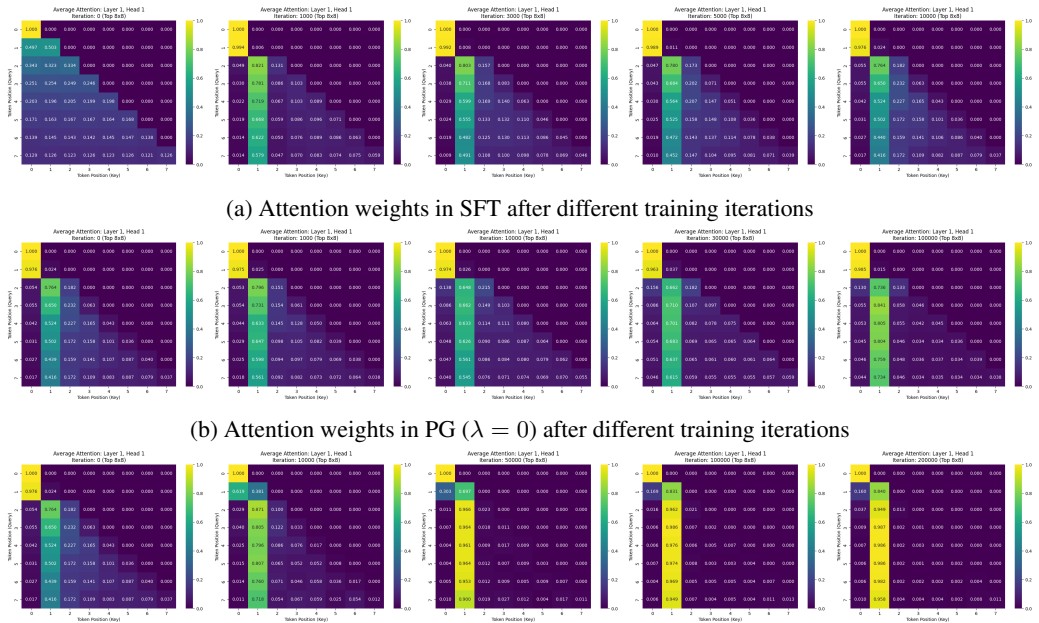

(a) Attention weights in SFT after different training iterations

(b) Attention weights in PG ($\lambda = 0$) after different training iterations

(c) Attention weights in Q-learning (Process Reward) after different training iterations

Figure 5: Empirical validation that the trained one-layer one-head transformer acts as a function of the target node and the current node. The visualization of attention maps across SFT, PG, and Q-learning training shows a consistent, strong focus on the target node (token position 1).

## G ADDITIONAL EXPERIMENTAL RESULTS

### G.1 VALIDATION OF LEARNED ATTENTION

This subsection visualizes the evolution of attention maps during training for SFT, PG, and Q-learning. Our analysis first focuses on a one-layer, one-head transformer at various training steps (Figure 5). For each model, we compute the average attention weight over the SFT training dataset $\mathcal{D}^{\text{SFT}}$.

Specifically, the weight at position $i$ in row $k$ (where $i \leq k$) represents the average attention the model assigns to the $i$-th token when predicting the $(k+1)$-th token, averaged over all paths in $\mathcal{D}^{\text{SFT}}$ longer than $k + 1$. Since the underlying graph has 100 sparse nodes, path lengths in $\mathcal{D}^{\text{SFT}}$ are generally short; consequently, we display only the first 8 rows.

Our visualizations reveal that during SFT, the transformer allocates most attention to the target node. Combined with the residual connections, which allow access to the current node's information, this suggests that the model learns to predict the next node based primarily on the target and current nodes. This empirical finding aligns with the results of Wang et al. (2024b). Another interesting phenomenon in SFT is that the attention weight on the target node quickly peaks and then gradually decreases, while remaining dominant. We hypothesize that this may be due to overfitting on $\mathcal{D}^{\text{SFT}}$, leading the model to develop auxiliary prediction strategies. This overfitting could also explain the decreasing generalization performance of SFT observed in our experiments.

In contrast, for both PG and Q-learning, the attention on the target node increases throughout training. In Q-learning, the final attention weight on the target node exceeds 95%, making it the closest to the conditions outlined in Assumption 3.1.

To further validate our findings, we extend the analysis to a two-layer, one-head transformer. As shown in Figure 6, which displays the attention map averaged over $\mathcal{D}^{\text{SFT}}$, both layers predominantly attend to the target and current nodes. This pattern strongly supports our assumption that the transformer operates as a function of the target and current nodes, confirming the consistency of this behavior across model architectures.

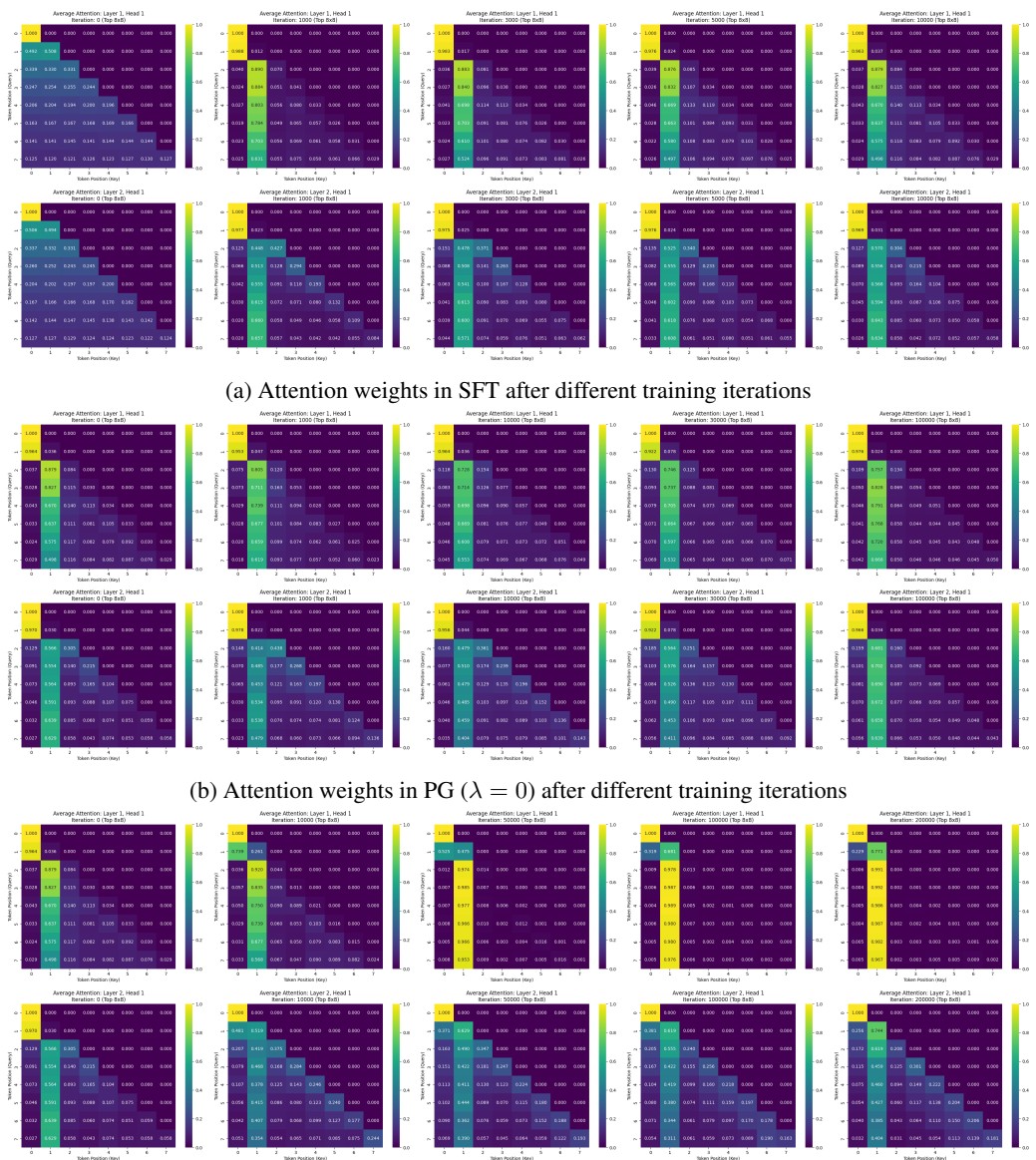

(a) Attention weights in SFT after different training iterations

(b) Attention weights in PG ($\lambda = 0$) after different training iterations

(c) Attention weights in Q-learning (Process Reward) after different training iterations

Figure 6: Empirical validation that the trained two-layer one-head transformer acts as a function of the target and current nodes.

## G.2 EXPERIMENTS ON ERDŐS-RÉNYI GRAPHS

Beyond the setup in Section 2, we conduct additional experiments to compare RL methods with SFT. The graph construction and initial SFT stage remain unchanged. After SFT, we split all **reachable pairs** into an RL training set $D_{\text{RL-Train}}$ and an RL test set $D_{\text{RL-Test}}$. This yields four intersections: $D_{\text{Train2Train}} := D_{\text{Train}} \cap D_{\text{RL-Train}}$, $D_{\text{Train2Test}} := D_{\text{Train}} \cap D_{\text{RL-Test}}$, $D_{\text{Test2Train}} := D_{\text{Test}} \cap D_{\text{RL-Train}}$, and $D_{\text{Test2Test}} := D_{\text{Test}} \cap D_{\text{RL-Test}}$.

During the RL process, the model generates paths for pairs in $D_{\text{RL-Train}}$ and receives reward signals. The main difference between this setup and that in Section 2 is that the RL training set now contains new pairs that were unseen during SFT ($D_{\text{Test2Train}}$). Therefore, the initial model is not perfect on these new training pairs. Additionally, some pairs from the SFT training set are not used for RL training ($D_{\text{Train2Test}}$), which allows us to measure the extent of forgetting. We consider the same

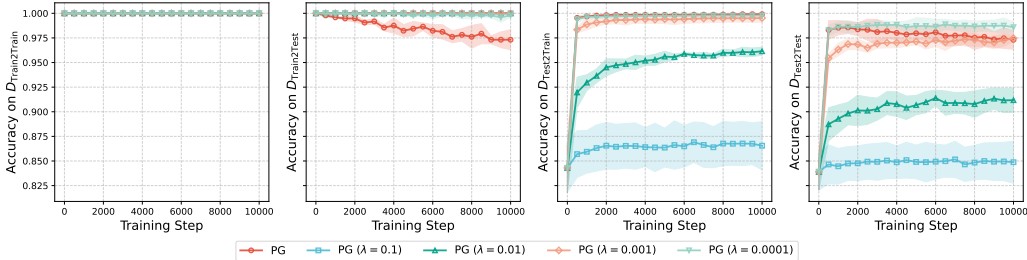

Figure 7: The test accuracy of PG with different KL coefficients on four data splits after fine-tuning the SFT model on $D_{\text{RL-Train}}$. All accuracies are evaluated with greedy decoding.

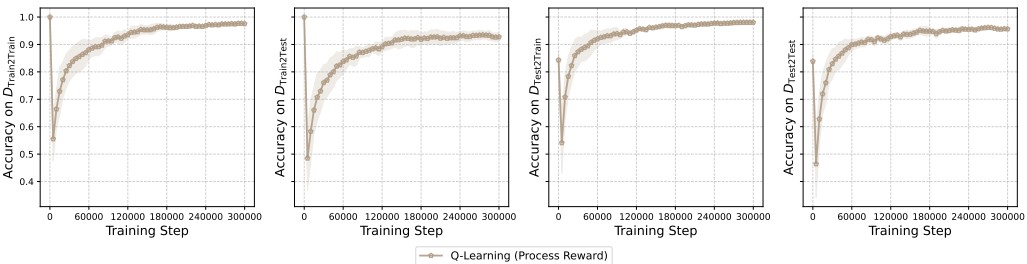

Figure 8: The test accuracy of Q-learning on four data splits after fine-tuning the SFT model on $D_{\text{RL-Train}}$. All accuracies are evaluated with greedy decoding.

RL algorithms introduced in Section 2: PG and Q-learning, whose training curves are presented in Figures 7 and 8, respectively. All accuracies are evaluated using greedy decoding.

From Figure 7, we observe the opposing effects of KL regularization on $D_{\text{Train2Test}}$ and $D_{\text{Test2Train}}$. PG without KL regularization ($\lambda = 0$) and less regularized PG ($\lambda = 0.0001$) achieve significantly higher accuracy on $D_{\text{Test2Train}}$. Stronger KL regularization hinders the model's ability to learn new pairs, which aligns with **Takeaway 4**: KL regularization reduces training accuracy. Conversely, PG without KL regularization ($\lambda = 0$) tends to overfit the training data and exhibits continual forgetting of previous knowledge learned during SFT. Results on $D_{\text{Test2Test}}$ further demonstrate that overly strong KL regularization can hinder PG's improvement. Among all settings, $\lambda = 10^{-4}$ achieves the best balance, indicating that a well-chosen KL weight can improve generalization with minimal sacrifice in training accuracy.

Compared to PG and the performance observed in Section 5, Q-learning exhibits slower convergence in this setting. One possible explanation is that the initial model performs poorly on the new training pairs, generating more failure cases and causing stronger "re-instantiation."

### G.3 EXPERIMENTS ON GRAPH REPRESENTED FOR BLOCKSWORLD

We also run experiments on Blocksworld (Valmeekam et al., 2023a), a benchmark for evaluating LLM planning ability (Kambhampati et al., 2024). The environment consists of blocks stacked on a table, and the goal is to rearrange the blocks from an initial configuration to a target configuration using a sequence of actions. We model this into a path-finding task, in which each configuration is a node in a graph, and an edge connects two nodes if one configuration can be transformed into the other by a single valid action, such as moving a block from one stack to another. We consider Blocksworld with four blocks and construct an undirected graph with 73 nodes: 24 configurations of a single stack of four blocks, 24 configurations with three blocks in one stack and one block on the table, 12 configurations with two stacks of two blocks, 12 configurations with one stack of two blocks and two blocks on the table, and one configuration with all blocks on the table.

Since accuracy comparison is not our focus, all node pairs are used for SFT training. The SFT dataset contains 50,000 paths sampled from the graph, with source and target nodes drawn uniformly from the 73 nodes. During RL training, the model generates paths for, and is updated on, all

node pairs. We use policy gradient and Q-learning as introduced in Section 2. After training, we evaluate the learned weights using the metric of Wang et al. (2024b), which measures the model's understanding of graph adjacency. As shown in Figure 1, with fixed training data, SFT may not learn the complete adjacency very well. In contrast, both PG and Q-learning improve the learned adjacency. In particular, Q-learning nearly recovers the complete adjacency, consistent with the results in Section 5.

