# OpenReview forum: "Benefits and Pitfalls of Reinforcement Learning for Language Model Planning: A Theoretical Perspective"
_ICLR.cc/2026/Conference — ICLR 2026 Poster_

### Official Review · Reviewer_BBiB · 2025-10-18

**Soundness:** 3
**Presentation:** 3
**Contribution:** 3
**Rating:** 8
**Confidence:** 3

**Summary:**

This paper provides a theoretical analysis of reinforcement learning (RL) methods, particularly policy gradient (PG) and Q-learning, for planning in large language models (LLMs). It builds on recent empirical works showing that RL enhances reasoning and planning beyond supervised fine-tuning (SFT), and offers a rigorous mathematical explanation of why and how that happens.

The authors adopt a graph-based abstraction of planning, where path-finding over a directed graph represents multi-step reasoning or decision-making (e.g., tool use, theorem proving, or task sequencing). They theoretically and empirically analyze three regimes 1) SFT: Learns co-occurrence-based correlations without transitive reasoning, leading to memorization rather than true planning; 2) Policy Gradient (PG): Outperforms SFT via exploration-based data augmentation, but suffers from diversity collapse (reduced output variety even after convergence). KL regularization mitigates this but can hurt accuracy; 3) Q-Learning: Addresses PG’s pitfalls by enabling off-policy learning and preserving output diversity at convergence—but risks reward hacking unless process rewards (stepwise feedback) are used instead of binary outcome rewards. The authors validate these behaviors using both synthetic Erdos–Rényi graphs and the Blocksworld planning benchmark

**Strengths:**

- This paper has a strong theoretical foundation: The paper provides a clean, formal framework for analyzing planning in LLMs, translating abstract reasoning tasks into tractable graph-based path-planning problems.

- Novel Theoretical Insights into RL for LLMs: Identifies diversity collapse in PG as a fundamental phenomenon, theoretically proving that entropy monotonically decreases after convergence—a widely observed empirical issue in RLHF and GRPO models. It reveals that Q-learning avoids this trap by updating state-action values rather than direct log-probabilities, preserving diversity across valid next actions. As well, it formalizes the reward hacking issue and demonstrates how process rewards eliminate it.

- Empirical Validation with Clear Visual Evidence: Figure 1 visualizes adjacency reconstruction in Blocksworld—showing how SFT fails to recover the graph structure while PG and Q-learning succeed. Then, the paper connects theory to practice via Figures 2–4. Figure 2 shows how KL regularization balances accuracy and diversity in PG. Figure 3 demonstrates Q-learning’s superior test accuracy and stability. Figure 4 draws heatmaps of logits reveal that feasible transitions converge to uniform values (confirming theoretical diversity preservation).

- Clear Conceptual Takeaways: The paper summarizes each section with concise takeaway statements, synthesizing intuition for both theoretical and empirical results. This makes it approachable despite heavy mathematical content.

**Weaknesses:**

- Limited Practical Scope: The paper primarily focuses on simplified models (one-layer, single-head transformers and synthetic graphs). While conceptually illuminating, the connection to real-world LLM training (e.g., PPO/GRPO pipelines) remains somewhat indirect. It has no experiments on large-scale reasoning benchmarks (e.g., MATH, GSM8K, or ALPINE) are included to show external validity.

- Simplistic Reward Modeling: The outcome vs. process reward distinction is theoretically elegant but lacks discussion on how these correspond to real LLM reward signals (e.g., chain-of-thought correctness, factual consistency). Furthermore, reward hacking is discussed in the graph setting but not demonstrated empirically in language tasks.

- Empirical Validation is Narrow: All empirical results use synthetic environments. If authors can have results on real text-based reasoning or tool-use tasks, then the statements would be much more strengthen that the same dynamics hold for modern RL-tuned LLMs.

- Absence of Theoretical Analysis for PPO/GRPO: Although Section F shows PPO is mathematically equivalent to unclipped PG under stop-gradient treatment, this misses important effects like clipping or value normalization used in real systems.

**Questions:**

- How do the diversity collapse dynamics extend to multi-step, natural language reasoning, where outputs are discrete tokens instead of graph edges?

- Can Q-learning be feasibly integrated with large transformer-based LLMs at scale? Are there existing implementations (e.g., in VeRL or R1) that align with the off-policy formulation analyzed here?

- Would adding entropy regularization or temperature annealing alter Theorem 4.3’s diversity-collapse dynamics?

- How sensitive are the conclusions to the assumption of persistent exploration? Could adaptive exploration decay or real-world sampling noise violate this condition?

- Could you relate process rewards to verifiable subgoal rewards used in reasoning tasks (e.g., partial correctness of intermediate steps)?

---

> ### Author Response · Authors · 2025-11-19
>
> We thank the reviewer for the thoughtful comments. Our responses are provided below.
>
> **Weaknesses 1&3**
>
> Thank you for pointing this out. Our primary goal in this paper is to develop theoretical understanding of RL in LLM training. The small-scale experiments are designed as controlled environments where our theoretical findings can be directly observed without confounding factors.
>
> We fully agree that validating these theoretical insights on larger models and more complex environments is important. However, such large-scale validation requires substantial computational resources and additional engineering considerations, which falls outside the scope of this theoretical paper. We view this as a valuable and promising direction for future work.
>
> **Weakness 2**
>
> In language tasks, the outcome reward signal is indeed straightforward: it simply checks whether the entire chain-of-thought is correct, i.e., whether the final answer and all reasoning steps are jointly valid.
>
> In contrast, the process reward is more complicated to define. One would need to evaluate whether each intermediate step is a valid transition, e.g., checking whether one node in the chain can logically lead to the next, and penalize the model immediately whenever a wrong intermediate step is produced.
>
> Regarding “reward hacking" in real language tasks, we believe a full empirical investigation is beyond the scope of this paper. We view it as an interesting direction for future work.
>
> **Weakness 4**
>
> Value normalization is mainly used to reduce variance, while keeping the expectation of the gradient unchanged. Therefore, it does not affect results such as Theorems 4.2 and 4.4. It would, however, modify Theorem 4.3: once the training accuracy reaches 100%, the normalized advantages become zero and the model stops updating. Nevertheles, in our experiments, diversity keeps decreasing even when the training accuracy is below 100%, so we believe that value normalization will not completely resolve the diversity-collapse phenomenon.
>
> As for clipping, it reduces variance but makes the gradient estimate biased, which substantially complicates the analysis. A full treatment of clipping is beyond the scope of this paper.
>
>
> **Question 1**
>
> As long as the reward signal remains 0/1 and the model is sufficiently expressive, we believe that the diversity collapse phenomenon will still occur in multi-step natural language reasoning. In particular, at any prefix, the logits of the correct next tokens follow the same dynamics described in Theorems 4.2 and 4.3.
>
> **Question 2**
>
> Designing an efficient and effective reward signal for large transformer-based LLMs remains challenging. Nevertheless, with a proper reward signal, we believe that Q-learning can be integrated into large LLM training and could be as efficient as policy gradient–based methods.
>
> There do exist recent attempts that apply Q-learning–style updates to transformer-based LLMs, such as [Ref A]. However, these methods do not align with the off-policy formulation analyzed in our paper, and therefore fall outside the specific theoretical regime we study.
>
> [Ref A] He H, Ye Y, Cai Q, et al. Random Policy Valuation is Enough for LLM Reasoning with Verifiable Rewards[J]. arXiv preprint arXiv:2509.24981, 2025.
>
> **Question 3**
>
> Adding entropy regularization is equivalent to applying KL regularization with a uniformly random base model. This can help maintain diversity, but may also lead to worse performance, since the base model in this case is very weak. As for temperature annealing, current practice typically decreases the temperature to below 1 toward the end of training, which would only further strengthen the diversity collapse phenomenon.
>
> **Question 4**
>
> Persistent exploration is very important in our analysis in Section 5. In fact, we believe that the convergence speed depends on the lowest exploration probability. If the exploration probability decays to 0, or if real-world sampling noise causes the model to never output certain tokens, then the persistent-exploration condition is violated, and our analysis would no longer apply.
>
> **Question 5**
>
> The process reward in Eq (3) can be viewed as a type of verifiable subgoal reward. In Eq (3), the process reward reflects local adjacency—it checks whether the next node is directly reachable from the current node. If a subgoal reward only verifies that the next state is reachable from the current state, then it is equivalent to the process reward. However, if the subgoal reward evaluates whether the next state can eventually reach the target state, then the two differ: the process reward is purely local, while the latter incorporates global reachability.

---

### Official Review · Reviewer_hAiC · 2025-11-01

**Soundness:** 3
**Presentation:** 4
**Contribution:** 3
**Rating:** 6
**Confidence:** 3

**Summary:**

This paper provides a theoretical comparison of different training paradigms for enabling language models to learn to plan—namely Supervised Fine-Tuning (SFT), Policy Gradient (PG), and Q-learning—within a controlled graph-based path planning setting.

The authors first show that SFT tends to memorize observed co-occurrences limiting generalization. They then demonstrate that PG improves over SFT despite having a similar objective, because its on-policy data generation introduces exploration, effectively augmenting the training set. However, they identify a diversity collapse in PG, where the model’s output variety diminishes even after achieving perfect training accuracy—a problem that KL regularization can mitigate at the cost of lower accuracy.

Finally, the authors analyze Q-learning, showing that it faces convergence issues and detail a form of “reward hacking” when relying solely on final outcome rewards, but that these problems can be addressed by introducing a process rewards to supervise intermediate steps. Under this setup, Q-learning naturally preserves output diversity and supports off-policy learning—unlike PG.

All theoretical predictions are validated empirically on a small Blocksworld planning environment using a one-layer Transformer.

**Strengths:**

- The paper is well written and clearly presents the theoretical analyses, making it easy to follow the derivations and main arguments.

- The controlled graph-based path planning framework provides a clean setting to extensively study the differences between SFT, PG, and Q-learning for planning with language models.

- The theoretical claims are relevant to the real world challenges of training LLM to plan.

**Weaknesses:**

**Lack of empirical validation at scale**
While the theoretical analyses are thorough and insightful, the empirical validation is limited to a small Blocksworld environment with a one-layer Transformer. It remains unclear how well the theoretical findings translate to larger, more complex models and real-world planning tasks. Additional experiments on larger datasets and models would strengthen the practical relevance of the results. Given the clearness of the theoretical contributions, it seems that it would be straightforward to validate them on larger models/environments.

**Figure 1 description**
Could you add more explanation to Figure 1? You only mention that SFT performs worse, but it’s not very clear how this is shown in the figure. For example, it seems that PG and Q-learning tend to hallucinate more than SFT — is that correct? I’m referring to the strong diagonal from bottom-left to top-right, which appears high for PG and Q-learning but low for SFT (as in the dataset). Also, Q-learning seems to produce a large bright block on the right — what does that correspond to?

**Lack of empirical validation for important assumption**
Theoretical analyses throughout the paper hinge on Assumption 3.1 — motivated by Wang et al. (2024b) — that the model’s logits depend only on the current and target nodes. However, this assumption is adopted without further empirical verification. The paper does not test whether real Transformer models indeed satisfy this reduced dependency, leaving open whether the derived results and takeaways accurately reflect practical LLM behavior.

**Questions:**

- Your theoretical results rely on **Assumption 3.1**, which posits that the model's logits for the next token depend only on the current node $u_m$ and the target node $u_2$, effectively neglecting the influence of the preceding path and initial state.
Could you discuss how your analysis would change if this assumption were relaxed—for instance, if the model's predictions also depended on earlier nodes in the trajectory (the full prefix $u_1, \ldots, u_m$)?
In particular, would the main results (e.g., Theorem 3.1’s co-occurrence characterization or the PG/Q-learning convergence properties) still hold qualitatively, or would new interaction terms or dependencies appear in the gradient dynamics?

- In your formulation, RL exploration appears to operate only over paths between fixed source–target pairs sampled from $D_{\text{Train}}$. Have you considered or analyzed the possibility of *exploring over source–target pairs themselves* (e.g., dynamically selecting or expanding the set of training pairs)? How might such goal-level exploration affect the generalization and theoretical results you present?

- In Figure 2, PG ($\lambda = 0$) achieves its highest test accuracy early in training before diversity collapse occurs, while KL regularization later limits accuracy gains. Wouldn't *early stopping* at this peak be a simpler and potentially more effective strategy than adding KL regularization, since it avoids both collapse and the regularization-induced accuracy loss? Did the authors consider this comparison?

- **How specific are your findings to transformer architectures?**
Your theoretical analysis and experiments use a one-layer Transformer as the backbone. How would the conclusions change for other model classes such as RNNs (e.g., LSTMs), state-space models (SSMs), or diffusion-based architectures? Do your results rely on properties unique to transformers (e.g., attention structure), or do they extend to other sequential models?

-**In real-world applications, process-level reward information is often unavailable.**
Given that your analysis shows Q-learning may fail to converge under outcome-only rewards, how could this limitation be addressed when such intermediate reward signals are not accessible?

---

> ### Author Response · Authors · 2025-11-19
> **Official Comment by Authors (1/2)**
>
> We really appreciate your insightful conceptual questions. Below are our responses.
>
> **Weakness 1**
>
> Thank you for pointing this out. Our primary goal in this paper is to develop theoretical understanding of RL in LLM training. The small-scale experiments are designed as controlled environments where our theoretical findings can be directly observed without confounding factors.
>
> We fully agree that validating these theoretical insights on larger models and more complex environments is important. However, such large-scale validation requires substantial computational resources and additional engineering considerations, which falls outside the scope of this theoretical paper. We view this as a valuable and promising direction for future work.
>
> **Weakness 2**
>
> Due to the presence of nonlinear layers, the brighter entries in the adjacency matrix cannot be directly interpreted as "hallucinations". Our intention in Figure 1 is not to draw conclusions from absolute brightness values, but from structural patterns of the learned adjacency. What we want to highlight is that both PG and Q-learning produce more coherent adjacency structures, particularly for transitions that do not frequently appear in the SFT dataset.
>
> In fact, Figure 1 is only intended as an illustrative motivation, the main conclusions rely on the subsequent theoretical analysis and controlled experiments.
>
> **Weakness 3**
>
> We agree that empirical validation is crucial. To address this, we have included a new analysis in Appendix G.1 that directly tests Assumption 3.1. We plot the evolution of attention maps across training, demonstrating that the models, especially Q-learning, learn to rely almost exclusively on the target node (with $>95\%$ attention weight), while the current node's information is accessible via residual connections. This confirms that the core dependency in our theoretical analyses is indeed reflected in the practical behavior of the transformers. This observation aligns with prior results, e.g., (Wang et al. 2024b). In addition, the attention to the goal states and recent states has been found in LLAMA-7B when applied to the planning problems [Ref A].
>
> [Ref A] Tianyi Men et al., Unlocking the Future: Exploring Look-Ahead Planning Mechanistic Interpretability in Large Language Models, EMNLP 2024.
>
> **Question 1**
>
> At a high level, the model naturally learns to suppress the influence of the earlier prefix, since those tokens do not provide useful information for predicting the next node. As training converges, their contribution becomes limited.
>
> Moreover, Assumption 3.1 is validated by our empirical observation that the model’s attention largely concentrates on the target node: in our experiments, the next node almost only depends on the current node $u_m$ and the target node $u_{target}$, while the influence of earlier nodes in the prefix is much weaker. That is, even if the assumption were relaxed, the transformer empirically converges to behavior consistent with Assumption 3.1. For this reason, we do not explicitly model the dependence on the rest of the prefix in our main analysis.
>
> **Question 2**
>
> Current RL training setups for LLMs typically use fixed questions. To reflect this practice, we also fix the source–target pairs during RL training.
>
> However, we believe that adaptively selecting source–target pairs would not change our theoretical results. On the one hand, the model may indeed produce new paths on newly introduced source–target pairs through exploration-driven data generation. On the other hand, the diversity-collapse phenomenon would still arise at convergence. Thus, the key mechanisms analyzed in our theory remain unaffected.
>
> **Question 3**
>
> Our goal in this work is to provide a complete timeline view of the RL training process: PG initially benefits from exploration-driven data generation, but eventually suffers from diversity collapse. Optimizing the training procedure, such as applying early stopping, is not the main focus of our paper. In fact, our analysis indeed provides a theoretical foundation for early stopping: it explains why PG achieves peak performance before collapse and why stopping at that point can be beneficial.

---

> ### Author Response · Authors · 2025-11-19
> **Official Comment by Authors (2/2)**
>
> **Question 4**
>
> As noted, the key requirement in our analysis is Assumption 3.1, together with the specific form of the loss function we study. If RNNs or SSMs satisfy this assumption and adopt the same loss formulation, then our analysis would apply to them as well.
>
> We next examine whether Assumption 3.1 holds for other backbones. As RNNs and SSMs do not have attention, we probe the last-layer features for the target node, current node, and other nodes (baseline). The linear probing accuracies are recorded in the following table. Our findings are: (1) For Transformer and LSTM, the next-node prediction depends primarily on the current and target nodes, which satisfies Assumption 3.1; (2) For Gated DeltaNet (SSM), dependence on the target node is weaker, suggesting that Assumption 3.1 may not hold for certain state-space models. This is an interesting direction for future investigation.
>
> $$
> \\begin{array}{|c|c|c|c|}
> \\hline
>  & \\text{Current Node} & \\text{Target Node} & \\text{Other nodes (AVG)} \\\\
> \\hline
> \\text{Transformer} & 100\\% & 98.3\\% & 30\\% \\\\
> \\hline
> \\text{LSTM (RNN)} & 99.8\\% & 99.3\\%   & 27.4\\% \\\\
> \\hline
> \\text{Gated DeltaNet (SSM)} & 99.8\\%  &  61.5\\% & 19.1\\% \\\\
> \\hline
> \\end{array}
> $$
>
> **Question 5**
>
> Process-level rewards are indeed often unavailable in real-world applications, and reward design becomes a major challenge for applying Q-learning in practice. This is an interesting direction for our future research.
>
> Moreover, in our setting, the process reward only penalizes illegal transitions (−1) and gives 0 for legal adjacency steps. Therefore, if the base model is sufficiently strong such that it almost never outputs illegal edges, then the process reward reduces to an outcome-only reward signal: all intermediate rewards are 0, and only the final successful transition yields +1. In this regime, the outcome reward becomes an effective approximation to the process reward.

---

### Official Review · Reviewer_q3k8 · 2025-11-02

**Soundness:** 1
**Presentation:** 1
**Contribution:** 2
**Rating:** 2
**Confidence:** 3

**Summary:**

This paper theoretically analyzes why LLMs work well for planning problems. They formulate the problem as  a graph where the prompt defines a start state and a target, and then the agent gets rewarded for generating a valid trajectory through the graph from start state to target.

They provide theoretical explanations for...
- why Policy gradient (PG) > SFT (exploration)
- why PG results in entropy collapse
- why KL regularization mitigates this
- why Q-learning on outcome rewards experiences "reward hacking" (their word for inaccurately optimistic Q-value estimates)
- How Q-learning with process rewards mitigates this

**Strengths:**

- Paper carefully cites where it extends prior work
- Claims are supported by experiments in Blocksworld, where they observe that KL regularization maintains output diversity during PG training and poor accuracy during Q-learning training with outcome rewards but good accuracy with process rewards.
- Graphs are clear and have error bars.

**Weaknesses:**

See the "Questions" section for more details, but at a high level, I think there are 3 areas for improvement:

- Make it clearer exactly how your theoretical analysis ties into real-world LLM setups. I think this could be made very explicit with a running example that you reference back to in each section (e.g. you could say something like "suppose you have a dataset consisting of math proofs. For each proof in the dataset, the starting state is [....]. The target is [.....]. Each state is [.....]. There is a valid transition between states if [....]..." Then when you introduce Q-learning you could say something like "The Q-function takes inputs in this form [...question, math proof up until step X...]. But when we optimize it, we see reward hacking [...predicting that invalid proofs are correct?...] Also discuss whether the assumptions you make when constructing graphs for your experiments are realistic assumptions that match data distributions in real natural language. (you have a line "a tool-use scenario can be modeled as identifying a valid call sequence within an API call graph" -- this is a good candidate to be fleshed out into a full example).

- The trajectory notation with u1 = start state, u2=target u3+ = actual trajectory was somewhat confusing to me. I'd suggest giving the start state and target different notation (e.g. s0, s_target) and then indexing the actual trajectory from 1.

- See my questions in the "Questions" section about the mismatch between the non-markovian setup in real LLMs and the markovian Q-learning setup.

- I think the phrase "reward hacking" is a bit misleading in this case. Typically this is used to refer to exploiting a misspecified external reward rather than having biased upward estimates of Q-value estimates within the policy.

**Questions:**

Disclaimer: it is quite possible that I have misunderstood some of the key arguments in this paper. If these could be clarified (both for myself and for future readers) my score could increase significantly.

Question 1 - could you clarify exactly what you mean by a "graph" in this setup? I think it was often ambiguous whether the paper is referring to a semantic graph (e.g. for a math proof, nodes are proof prefixes and an edge from A to B exists is B contains A, but with one more logically correct step added (or is it one valid next token added?)) or to a literal graph (vocabulary are nodes, graph is fully connected b/c any token can come after any other one).

I'm also very confused about whether we are assuming the graph is markovian or not. In section 2.1, it says that each vocabulary item is a word. However, whether a transition from one word to the next is valid depends not just on the current word, but on all prior words in the sequence. Furthermore autoregressive transformers look at sequence prefixes not just words in isolation. So is a node a single word, or a sentence prefix? If it's a single word, I don't think this matches how LLM RL is done in practice.

This confusion further makes me question the Theorem 5.1 results. It makes sense that you would converge to equal Q-values for all next states if it is possible to select k' across all possible next nodes. However if nodes are prefixes it's not clear to me if this still holds. My mental model is a problem where the start token is "Hello" and the model gets reward 1 if its next token is "World", but reward 0 for anything else ever. If a transformer is trained on a batch of trajectories with "Hello World"-> 1 and [anything else]-> 0 it is not clear to me the theorem still holds. If a sequence is [random, sequence, of, words....] then there is no possible transition from that to a sequence [Hello, World], so I don't think Q-learning would ever give this sequence > 0 reward.

Question 2 - I think some of the discussion about SFT vs PG was a bit unclear:
- The paper states that "SFT memorizes" whereas "PG generalizes". The claim that SFT memorizes makes sense and was well supported. However, the claim that "PG generalizes" seems at odds with the analysis showing a loss of diversity. It seems like PG is not truly generalizing, merely memorizing a small set of paths from start state to the solution. The Fig 2a results do show stronger test performance of RL methods, but the "Continual SFT" setup was not described, so I'm not sure what it means.
- There are 2 differences between SFT and PG: (a) PG has a notion of reward which SFT lacks (b) SFT is static whereas PG collects new trajectories online. The theoretical analysis focuses on the benefit of (b) but I think does not address (a).

---

> ### Author Response · Authors · 2025-11-19
> **Official Comment by Authors (1/2)**
>
> We thank the reviewer for the helpful comments. Here are our responses.
>
> **Weakness 1**
>
> Thank you for the suggestion. We agree that making the connection between our theoretical analysis and real-world LLM planning scenarios more explicit is very important. Below, we illustrate this connection using a concrete example: Blocksworld, a common benchmark for evaluating LLM planning capabilities (Valmeekam et al., 2023b). This example is also explained in our revised Section 2.1.
>
> Blocksworld consists of a set of colored blocks placed either on a table or stacked on top of each other. The goal is to plan a sequence of moves that transforms an initial arrangement (source state) into a target arrangement (target state). For example, suppose we have 4 blocks: Grey (G), Black (B), Red (R), and White (W). The source state is all blocks on the table (G|B|R|W), and the target state is a tower where Red is on Grey, Grey is on Black, and Black is on White (RGBW).
>
> Following Wang et al. (2024b), we abstract this scenario into a graph $G = (V, E)$. Each node in $V$ corresponds to a valid block arrangement (state). For example, Node 0 represents all blocks on the table (G|B|R|W), and Node 1 represents the state where W is on G while others remain on the table (WG|B|R). An edge connects two nodes if one can be reached from the other by exactly one valid move; for instance, Node 0 and Node 1 are connected because we can obtain Node 1 by placing W on G. A valid path is a sequence of nodes from the source to the target, where each transition follows a legal move, and each node is represented by one unique token.
>
> This abstraction aligns naturally with how we generate training data and apply reinforcement learning. For supervised fine-tuning (SFT), we sample a valid path from the graph between the source and target nodes as ground truth. In a natural language interface, this corresponds to instructions such as “Move White onto Grey, and get state (WG|B|R). Move Red onto Black, and get state (WG|RB). Move White onto table, and get state (RB|W|G).” For reinforcement learning (RL), the model is only given the source and target nodes and asked to generate a sequence of nodes. The reward function then checks whether the generated sequence constitutes a valid path.
>
> We deliberately abstract away natural language semantics to enable clean theoretical analysis while retaining the essential planning structure. This type of abstraction is commonly adopted in the literature (Wang et al., 2024b and [Ref A]).
>
> [Ref A] Hanlin Zhu et al.,  Emergence of Superposition: Unveiling the Training Dynamics of Chain of Continuous Thought, Arxiv 2025.09
>
> **Weakness 2**
>
> Thank you for pointing this out. In the revised version, we have adopted your suggestion by introducing distinct symbols for the start state ($u_{source}$) and target state ($u_{target}$), and by indexing the actual trajectory as $u_1,\cdots,u_m,\cdots$.
>
> **Weakness 4**
>
> Thank you for this helpful clarification. We agree that “reward hacking” is not the most accurate term for the phenomenon we study. In our setting, the outcome reward we use is exactly the original task objective. However, it leads to an incorrect Q-value estimation, which is not an exploitation of a misspecified external reward.
>
> To avoid confusion, we have removed the term “reward hacking” from the paper and now describe this effect as “Q-value bias”.
>
> **Question 1**
>
> Thank you for the question. We provide a full definition and example of the graph in our response to Weakness 1, which clarifies that the graph is a simple directed graph, rather than a semantic graph over tokens.
>
> Regarding the Markovian aspect: our RL formulation is modeled as an MDP, so the task is Markovian at the state level. In a simple directed graph, the validity of the next move depends only on the current node and the target node, not on the entire prefix. This is also supported by our experiments: although the transformer receives the full prefix as input, the attention maps (see revised Fig. 5 and Fig. 6 in Appendix G.1) show that it primarily focuses on the current node and the target node, indicating that the model can automatically identify and exploit this Markovian structure.
>
> Finally, for Theorem 5.1, the analysis is carried out over graph nodes rather than arbitrary token prefixes. Under the empirical observation that the transformer mainly attends to the current and target nodes, this assumption is reasonable, and the theorem follows accordingly.

---

> ### Author Response · Authors · 2025-11-19
> **Official Comment by Authors (2/2)**
>
> **Question 2.1**
>
> The statement "SFT memorizes whereas PG generalizes" is concluded by prior works, e.g., (Chu et al., 2025), and our paper aims to provide a theoretical explanation for this empirical phenomenon.
>
> Importantly, by "generalization" we specifically refer to the ability to learn new relationships (e.g., new reachabilities) that do not appear in the supervised dataset $D^{SFT}$. Under this definition, PG indeed "generalizes" due to its exploration-driven data generation.
>
> The reviewer correctly notes that PG reduces trajectory diversity. We agree, and our theory explicitly characterizes this collapse. However, reduced trajectory diversity does not contradict generalization in relationships: PG may eventually focus on a small set of trajectories, but those trajectories may rely on previously unseen transitions, which SFT cannot learn without exposure.
>
> Regarding Continual SFT, it refers to continuing to train the same base model on the same SFT dataset for the same number of total gradient steps as PG, serving as a fair comparison baseline that controls for training compute. This is explained in the first paragraph of Section 4.2.
>
> **Question 2.2**
>
> The SFT loss can be viewed as using an outcome reward, where all trajectories in the SFT dataset receive reward 1 simply because they are correct demonstrations. In Theorem 4.1, when PG is trained with a 0–1 reward, the trajectories with reward 0 do not contribute to the gradient. Thus, the effective learning signal for PG also comes only from reward-1 trajectories, which contain the same type of information as those in $D^{SFT}$.
>
> Therefore, the reward structure itself does not explain the performance difference between SFT and PG. What truly matters is (b): PG’s online rollouts allow the model to access new successful trajectories that do not exist in $D^{SFT}$, enabling new value updates that SFT fundamentally cannot obtain.

---

### Meta-Review · Area_Chair_dbeo · 2026-01-09

**Summary:**

This paper theoretically analyzes why LLMs work well for planning problems. They formulate the problem as a graph where the prompt defines a start state and a target, and then the agent gets rewarded for generating a valid trajectory through the graph from start state to target.

They provide theoretical explanations for a set of interesting phenomena:
why Policy gradient (PG) > SFT (exploration)
why PG results in entropy collapse
why KL regularization mitigates this
why Q-learning on outcome rewards experiences "reward hacking" (their word for inaccurately optimistic Q-value estimates)
How Q-learning with process rewards mitigates this

The reviewers generally appreciated the paper, with a few concerns:
1. Connection of assumptions/setup to real LLMs.
2. Lack of empirical validation.
3. Confusing notation.
4. Definition of a graph and whether it is Markovian
5. Strong assumptions that may not hold in reality.
6. Analysis only for PG not for PPO/GRPO.

**Reviewer Concerns:**

The reviewers did a good job of addressing reviewer concerns in the rebuttal. Reviewer q3k8 had some conceptual misunderstandings that were well resolved by the authors. The authors added a bunch of clarifications which help with the understandability of the paper. They also added an experiment showing that their assumption 3.1 holds (in Appendix G.1). I think that this addresses a lot of the concerns raised by the reviewers. While the analysis is somewhat simplified, as analysis often is - it is still valuable.

**Reviewer Scores:**

q3k8 would have raised to a 4/5
hAiC either remaining the same or raising by 1
BBiB remaining the same (already accept)

Overall I think the paper goes over to an accept.

---

### Decision · Program_Chairs · 2026-01-26

Accept (Poster)